RESEARCH COMMUNICATION

# Human pancreatic cancer cell exosomes, but not human normal cell exosomes, act as an initiator in cell transformation

Karoliina Stefanius[1,2], Kelly Servage[1,2], Marcela de Souza Santos[1], Hillery Fields Gray[1,2], Jason E Toombs[3], Suneeta Chimalapati[1,2], Min S Kim[4], Venkat S Malladi[4], Rolf Brekken[3,5], Kim Orth[1,2,6]*

[1]Department of Molecular Biology, University of Texas Southwestern Medical Center, Dallas, United States; [2]Howard Hughes Medical Institute, University of Texas Southwestern Medical Center, Dallas, United States; [3]Division of Surgical Oncology, Department of Surgery, Hamon Center for Therapeutic Oncology Research, University of Texas Southwestern Medical Center, Dallas, United States; [4]Department of Bioinformatics, University of Texas Southwestern Medical Center, Dallas, United States; [5]Department of Pharmacology, University of Texas Southwestern Medical Center, Dallas, United States; [6]Department of Biochemistry, University of Texas Southwestern Medical Center, Dallas, United States

**Abstract** Cancer evolves through a multistep process that occurs by the temporal accumulation of genetic mutations. Tumor-derived exosomes are emerging contributors to tumorigenesis. To understand how exosomes might contribute to cell transformation, we utilized the classic two-step NIH/3T3 cell transformation assay and observed that exosomes isolated from pancreatic cancer cells, but not normal human cells, can initiate malignant cell transformation and these transformed cells formed tumors in vivo. However, cancer cell exosomes are unable to transform cells alone or to act as a promoter of cell transformation. Utilizing proteomics and exome sequencing, we discovered cancer cell exosomes act as an initiator by inducing random mutations in recipient cells. Cells from the pool of randomly mutated cells are driven to transformation by a classic promoter resulting in foci, each of which encode a unique genetic profile. Our studies describe a novel molecular understanding of how cancer cell exosomes contribute to cell transformation.
**Editorial note:** This article has been through an editorial process in which the authors decide how to respond to the issues raised during peer review. The Reviewing Editor's assessment is that major issues remain unresolved (see decision letter).
DOI: https://doi.org/10.7554/eLife.40226.001

*For correspondence:
kim.orth@utsouthwestern.edu

## Introduction

Within the tumor microenvironment, a dynamic molecular communication between tumor and surrounding stromal cells is a well-recognized feature of cancer progression (*Salvatore et al., 2017*; *Kahlert and Kalluri, 2013*). The conversation at the primary tumor site, as well as at distant locations, is mediated through many secreted factors including exosomes; small (30–150 nm) secreted extracellular vesicles shed by normal and malignant cells (*Kalluri, 2016*; *Costa-Silva et al., 2015*; *Stahl and Raposo, 2018*; *Colombo et al., 2014*; *Willms et al., 2016*). Exosomes and their mechanism(s) of biogenesis and function has emerged as a promising, yet controversial, field of research. Based on multi-omic studies, exosomes are known to carry heterogeneous cargo composed of proteins, metabolites, genetic material (DNA and microRNAs), and lipids (*Willms et al., 2016*;

*Demory Beckler et al., 2013*; *Kowal et al., 2016*; *Kalluri and LeBleu, 2016*; *Melo et al., 2014*; *Smith et al., 2015*). They are selectively packaged and transferred into recipient cells acting as vehicles for intercellular communication in normal physiological and pathological conditions (*Hessvik and Llorente, 2018*). A growing body of evidence shows exosomes are crucial in shaping the local tumor microenvironment to promote cancer progression by advancing tumor metastasis (*Costa-Silva et al., 2015*; *Zhang et al., 2015*). Although there is a major emphasis on describing the function of exosomes in metastasis and interactions in the local tumor microenvironment, less effort has been invested in analyzing their specific contribution to transforming a normal cell into a malignant cell. It has been shown that exosomes can contribute to the transformation of nontumorigenic cells to form tumors (*Melo et al., 2014*; *Dai et al., 2018*; *Abdouh et al., 2017*; *Hamam et al., 2016*; *Antonyak et al., 2011*). However, a common feature among these studies is the use of cells that were either pre-exposed to transforming agents or treated with cancer patient sera or medium from cultured cancer cells, both of which contain cancer cell exosomes. Under these conditions, it is conceivable that other components, in addition to exosomes, are transforming cells.

Pancreatic cancer is a lethal metastatic disease that lacks efficient curative treatment. Even though there is continuing progress toward understanding the biology of pancreatic cancer, it remains one of the leading causes of cancer-related deaths in the world (*Vincent et al., 2011*). This is mainly due to the lack of effective treatments against its highly metastatic behavior and therefore, understanding the key mechanisms underlying its progression is needed. Mutations in the four pancreatic driver genes, *KRAS*, *CDKN2A*, *TP53*, and *SMAD4*, occur frequently in pancreatic ductal adenocarcinomas (PDAC) and are well described (*Giovannetti et al., 2017*). An activating mutation in the *KRAS* gene is present in 90% of cases (*Giovannetti et al., 2017*). Additionally, genetic heterogeneity and polyclonality have also been shown to be present in PDAC (*Giovannetti et al., 2017*). Together with the indications that cancer-cell-derived exosomes are emerging contributors to tumor promotion, we wanted to evaluate whether exosomes secreted by pancreatic cancer cells participate in a distinct role in the process of cell transformation.

Malignant transformation of a normal cell occurs in a stepwise fashion. Point mutations in the genome can result in the reprogramming of a normal cell to a less differentiated state that is receptive to additional genetic alterations resulting in uncontrolled growth and ultimately cancer. The classic two-stage in vitro cell transformation assay (CTA) is a tiered system for transformation that was created for screening potential carcinogenic factors (*Berwald and SACHS, 1963*; *Kakunaga, 1973*; *Sakai and Sato, 1989*). In this system, cells are first treated with a suspected carcinogen, called an *initiator*, such as the genotoxic carcinogen 3-MCA (3-methylcholanthrene). 3-MCA introduces random genetic changes in a pool of normal cells. Subsequently, these *initiated* cells are exposed to a *promoter*, such as TPA (12-O-tetradecanoylphorbol 13-acetate), which enhances proliferation in the *initiated* cells selectively, thus driving malignant transformation of the cells. The resulting transformed cells are observed as foci on a cell culture plate (*Sakai and Sato, 1989*; *Sasaki et al., 2012*). This reductionist approach provides sensitivity in detecting a wider range of *initiating* agents that may not show obvious transforming activity without a *promoter* (*Sakai and Sato, 1989*). Using this assay as a model system for malignant cell transformation, we assessed whether or not cancer-cell-derived exosomes could affect and/or potentially drive the transformation of a normal cell.

The results presented herein provide a detailed analysis of a previously unidentified molecular function of cancer cell exosomes for malignant cell transformation. We observe that exosomes derived from pancreatic cancer cells can act as an *initiator* but not as a *promoter* in the two-stage CTA leading to malignant cell transformation. By contrast, exosomes derived from normal pancreatic cells have no effect on the cell transformation process. Specifically, using this two-stage CTA, we observe over a three-day initiator step that a single treatment of cancer cell exosomes acts in the same manner as a single treatment of the chemical *initiator* 3-MCA. As *initiators* of cell transformation, they incorporate random molecular changes into DNA. These random mutations then set the stage for a *promoter* to induce transformation of cells to form foci. In addition, we show that the cancer cell exosome-initiated transformed cells form aggressive tumors when injected into mice. In-depth analysis of the transformed cells using a combination of proteomics and exome sequencing reveals distinct differences between healthy cells and transformed cells with key insights into how cancer cell exosomes may be working on recipient cells to contribute to transformation. Overall, this study uncovers a specific function of pancreatic cancer cell exosomes in the malignant cell transformation process.

## Results

### Exosome isolation, validation, and characterization

Exosomes were isolated using a combined ultrafiltration-ultracentrifugation protocol (described in detail in Materials and methods) from four pancreatic cancer cell lines, Capan-2, MIA PaCa-2, Panc-1, and BxPC-3, and two human normal cell lines, human pancreatic ductal epithelial cells (HPDE) and human primary dermal fibroblasts (*Adamczyk et al., 2011*). Of note, three of the cancer cell lines, Capan-2, MIA PaCa-2, and Panc-1, have oncogenic mutations in the *KRAS* gene, whereas BxPC-3 has wild-type *KRAS* (*Deer et al., 2010*). To confirm rigor and reproducibility, isolated exosomes from each cell type were characterized for the presence of common exosome-associated proteins using mass spectrometry and immunoblot analysis (*Figure 1*, *Figure 1—figure supplement 1*, *Figure 1—figure supplement 2*). Additionally, electron microscopy (TEM) and nanoparticle tracking analysis (NTA) were employed to analyze the morphology and size range of isolated exosomes (*Figure 1*, *Figure 1—figure supplement 1*) (*Willms et al., 2016*; *Kowal et al., 2016*; *Lötvall et al., 2014*; *Witwer et al., 2017*; *Théry et al., 2018*). An example of this characterization of exosomes is shown for Capan-2 exosomes in *Figure 1*. Immunoblot analysis was used to confirm the presence of expected exosomal marker proteins (CD63, Alix, and TSG101) as well as the absence of proteins not commonly found in exosomes (Calnexin, α-actinin, and HSP90) (*Figure 1A,B*). In addition to western blot analysis, mass spectrometry analysis of isolated exosomes confirms the presence of the top twenty most commonly found proteins in exosomes according to the ExoCarta database (*Figure 1—figure supplement 2*) (*Keerthikumar et al., 2016*). TEM images of exosomes isolated from Capan-2 cells show the expected round or cup-shaped morphology (*Willms et al., 2016*) and NTA shows the size distribution of exosomes centered on 91 nm with a mean size of 250.3 nm (*Figure 1C,D*).

### Pancreatic cancer cell exosomes function as an *initiator* in malignant cell transformation

To analyze if exosomes contribute to malignant cell transformation, we utilized the two-stage CTA as a model system for transformation. The CTA was performed with NIH/3T3 cells using an established chemical *initiator* and *promoter*, MCA and TPA, respectively (*Sakai and Sato, 1989*; *Alvarez et al., 2014*). As shown in *Figure 2A*, the complete assay is 42 days long and involves the treatment of NIH/3T3 cells with an *initiator* (3 days) followed by a *promoter* (2 weeks) before recovery (3 weeks). Successful cell transformation results in the formation of foci that are identified by defined criteria described in Materials and methods (*Figure 2B*). Cells that are untreated, treated with only an *initiator* (MCA), or only a *promoter* (TPA) show the formation of low levels of background foci (one foci/well on average) (*Figure 2C*, *Figure 2—source data 1*) (*Sasaki et al., 2012*). By contrast, cells treated with both MCA as the *initiator* and TPA as the *promoter* resulted in formation of 3–4 foci/well on average (*Figure 2C*, *Figure 2—source data 1*). Consistent with previous findings, treatment with both an *initiator* and *promoter* was required to observe increased cell transformation as evidenced by formation of an increased number of foci above background levels (*Sakai and Sato, 1989*).

To establish what role, if any, cancer-cell exosomes have on cell transformation, we assessed whether exosomes isolated from three pancreatic cancer cell lines, Capan-2, MIA PaCa-2, and Panc-1, which are known to carry oncogenic mutations in the *KRAS* gene, could act as an *initiator* and/or *promoter* in the CTA. NIH/3T3 cells were first treated with isolated exosomes from each of the three pancreatic cancer cell lines for the duration of the *initiation* and *promotion* steps (3-week treatment). This resulted in the formation of only background transformation activity, similar to what is observed with the untreated control (*Figure 2C*, *Figure 2—source data 1*). Next, NIH/3T3 cells were first treated with the *initiator* MCA and then subsequently treated with isolated cancer cell exosomes for the 2-week *promotion* period. For each of these three cancer cell exosome assays, again only background levels of foci were observed (*Figure 2C*, *Figure 2—source data 1*). However, when the cancer cell exosomes were tested as an *initiator* in combination with the *promoter* TPA, cell transformation was observed at similar levels as the chemical MCA/TPA treatment (3–4 foci/well) (*Figure 2C*, *Figure 2—source data 1*). Therefore, exosomes derived from three different pancreatic cancer cell lines (Capan-2, MIA PaCa-2 or Panc-1) can each function as an *initiator* in the CTA resulting in transformation of NIH/3T3 cells.

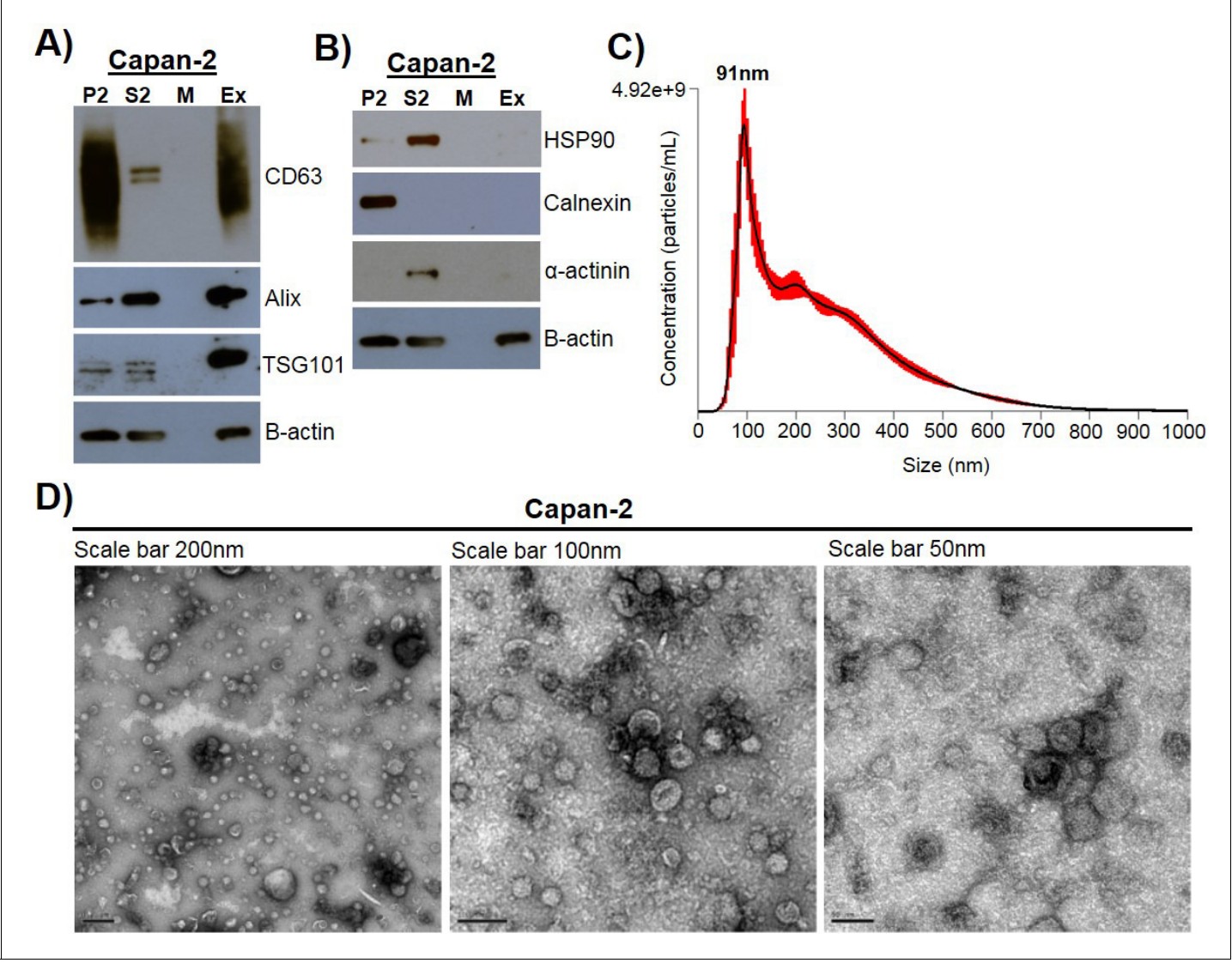

**Figure 1.** Exosome isolation, validation, and characterization. (A) Western blot analysis of common exosomal marker proteins CD63, Alix, and TSG10 found in exosomes isolated from Capan-2 cells. (B) Western blot analysis of proteins HSP90, Calnexin, and α-actinin, expected to be underrepresented in exosomes. Equivalent amounts of proteins from P2 (ER and mitochondria), S2 (cytoplasm), M (media), and Ex (exosome) fractions derived from the Capan-2 exosome isolation process were loaded into gel for the analysis. (C) Nanoparticle Tracking Analysis of 'crude' Capan-2 cell exosomes. Data represent average size per concentration (black line) ± standard error of the mean (red bars) of three measurements from one exosome preparation. Exosome size is centered on 91 nm with a mean size of 250.3 nm. Finite Track Length Analysis (FTLA) was used for size determination. (D) Representative TEM images of exosomes isolated from Capan-2 cells shown at three different scales confirm expected cup-shaped morphology of vesicles.

DOI: https://doi.org/10.7554/eLife.40226.002

The following source data and figure supplements are available for figure 1:

**Figure supplement 1.** Exosome isolation, validation, and characterization of additional cell types.
DOI: https://doi.org/10.7554/eLife.40226.003

**Figure supplement 2.** Twenty most common proteins found in exosomes according to ExoCarta database identified by mass spectrometry analysis in 'crude' Capan-2 exosome sample and 'pure' Capan-2 exosome sample (Fraction three from sucrose density gradient).
DOI: https://doi.org/10.7554/eLife.40226.004

**Figure supplement 2—source data 1.** Relates to *Figure 1—figure supplement 2*.
DOI: https://doi.org/10.7554/eLife.40226.005

**Figure supplement 2—source data 2.** Relates to *Figure 1—figure supplement 2*.
DOI: https://doi.org/10.7554/eLife.40226.006

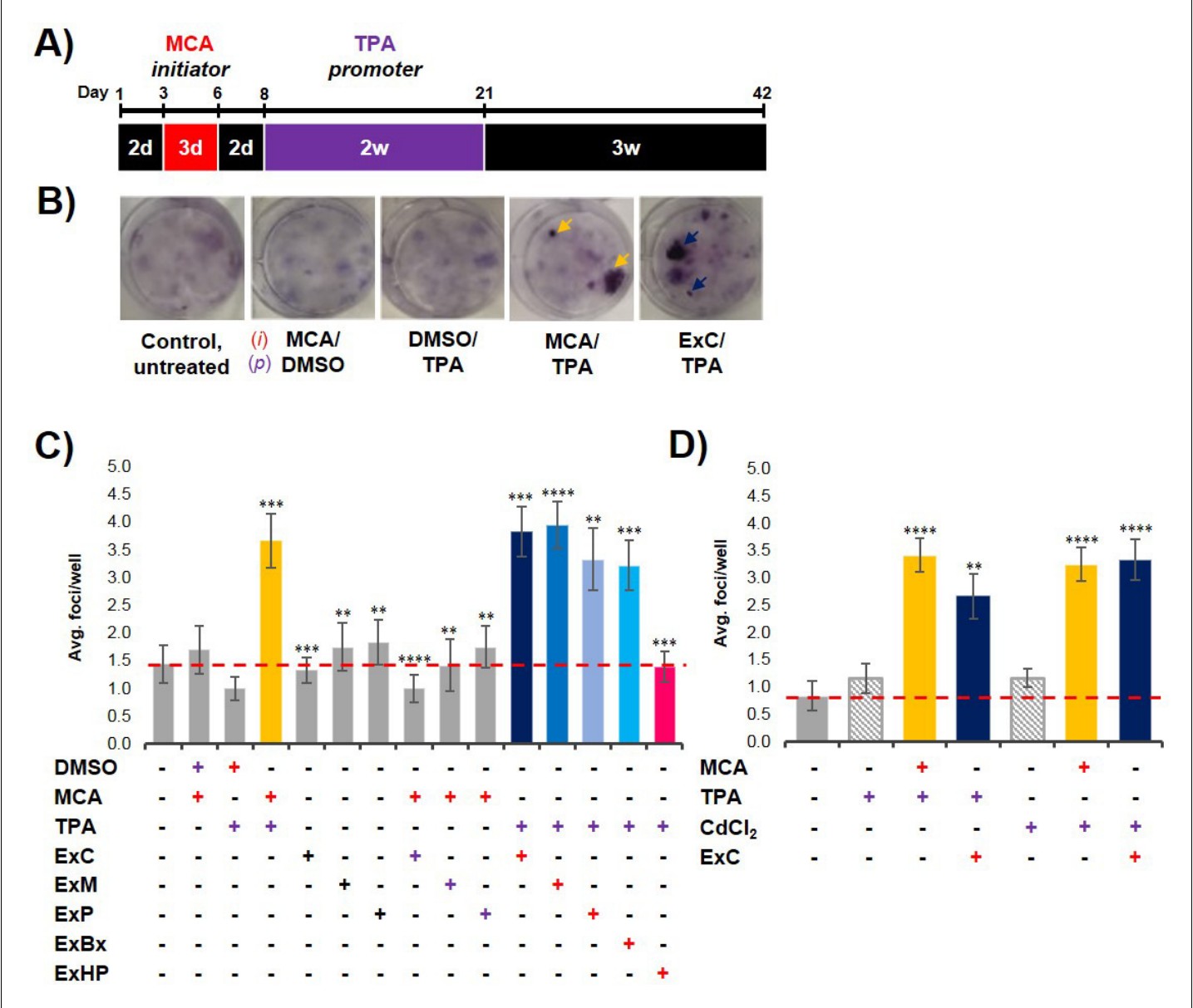

**Figure 2.** Pancreatic cancer cell exosomes function as an initiator in malignant cell transformation. (**A**) Two-stage cell transformation assay shown. NIH/3T3 cells were treated with a tumor initiator for 3 days (Days 3–6) and the tumor promoter for 2 weeks (Days 8–21). After 42 days, cells are fixed with methanol and stained with Crystal Violet for malignant foci counting. (**B**) Representative images of stained cells showing foci formation (arrows) from untreated cells and cells treated with MCA/DMSO, DMSO/TPA, MCA/TPA, or Capan-2 exosomes (ExC)/TPA (initiator/promoter). (**C**) Quantification of foci formed at the end of cell transformation assays. The average foci/well were determined via double-blind counting as described in Materials and methods. The red dashed line represents the established level of background foci present in untreated cells. Initiator/promoter treatments resulting in increased foci formation above background include MCA/TPA (p=0.008) and all cancer cell-derived exosomes: ExC/TPA (p=0.0002), ExM/TPA (p<0.0001), ExP/TPA (p=0.007), and ExBx/TPA (p=0.0003). Bars shown in gray represent controls that did not result in foci formation above background. Bar shown in pink shows results from normal cell (HPDE) exosome/TPA treatment (p=0.0004). (**D**) Quantification of foci formed after use of a different promoter, CdCl$_2$. CdCl$_2$ acts as a promoter leading to increased foci formation above background when used with the initiators MCA (p<0.0001) or Capan-2 exosomes (ExC) (p<0.0001). Asterisks indicate significant differences from either control treatment or MCA/TPA treatment as determined by unpaired, two-tailed t-test with Welch's correction (*p<0.05; **p<0.01; ***p<0.001; ****p<0.0001). Red (+)=initiator used; purple (+)=promoter used.

DOI: https://doi.org/10.7554/eLife.40226.007

The following source data and figure supplements are available for figure 2:

**Source data 1.** Relates to *Figure 2*.

DOI: https://doi.org/10.7554/eLife.40226.012

*Figure 2 continued on next page*

*Figure 2 continued*

**Source data 2.** Relates to *Figure 2*.
DOI: https://doi.org/10.7554/eLife.40226.013
**Figure supplement 1.** Characterization of initiation activity of pancreatic cancer cell exosomes.
DOI: https://doi.org/10.7554/eLife.40226.008
**Figure supplement 1—source data 1.** Relates to *Figure 2—figure supplement 1*.
DOI: https://doi.org/10.7554/eLife.40226.009
**Figure supplement 1—source data 2.** Relates to *Figure 2—figure supplement 1*.
DOI: https://doi.org/10.7554/eLife.40226.010
**Figure supplement 1—source data 3.** Relates to *Figure 2—figure supplement 1*.
DOI: https://doi.org/10.7554/eLife.40226.011

To assess whether this *initiator* activity is a general characteristic of exosomes from all cell types or a trait unique to exosomes derived from *KRAS* mutated pancreatic cancer cell lines, we repeated experiments with exosomes from three additional cell types: BxPC-3 cells, a pancreatic cancer cell line with WT *KRAS*, HPDE cells, a normal human cell line, and primary dermal fibroblasts. We observed that cancer cell exosomes isolated from WT *KRAS* BxPC-3 cells can act as an *initiator* of cell transformation, resulting in foci formation similar to the numbers observed with MCA/TPA treatment (*Figure 2C*, *Figure 2—source data 1*). However, normal cell exosomes isolated from HPDE cells or exosomes from primary fibroblasts were unable to induce cell transformation when used as an *initiator* in the CTA (*Figure 2C*, *Figure 2—source data 1*, *Figure 2—figure supplement 1A*, *Figure 2—figure supplement 1—source data 1*). Collectively, the results show that pancreatic cancer cell exosomes can act as an *initiator* in malignant cell transformation of NIH/3T3 cells, while exosomes isolated from normal pancreatic cells or primary fibroblasts cannot.

## Gradient purified exosomes contain *initiator* activity

Exosomes were isolated using an ultrafiltration-ultracentrifugation method and validated using a number of criteria (*Figure 1*). Our results demonstrate a specific function for cancer cell exosomes, but we used a protocol that is known to result in a preparation containing both exosomes and aggregated protein/nucleic acid contaminants. For this reason, we performed an additional purification step by floating exosomes onto a sucrose density gradient to obtain cleaner exosome preparation separated from contaminants (*Chiou and Ansel, 2016*). Characterization of these purified exosomes (Fraction 3) by NTA showed a size range centered on 67 nm with a mean size of 83.5 nm. Immunoblot analysis confirmed the presence of expected exosomal marker proteins CD63 and Alix. In addition, mass spectrometry analysis confirmed the presence of the top twenty most commonly found proteins in exosomes according to the ExoCarta database (*Figure 1—figure supplement 1*, *Figure 1—figure supplement 2*). The purified exosomes were then tested as an *initiator* with the *promoter* TPA in the transformation assay and results show that the population of 'pure' exosomes retain the ability to act as an *initiator* of cell transformation (*Figure 2—figure supplement 1B*, *Figure 2—figure supplement 1—source data 2*).

## Cancer cell exosome *initiator* activity is detected at low concentrations, requiring intact exosomes

Dose-response studies were performed using protein concentration as a normalization strategy to evaluate the amount of cancer cell exosomes needed to *initiate* cell transformation. As a standard concentration in each transformation assay, we used 80 ng/mL of proteins, corresponding to $7.0 \times 10^7$ particles/mL. Exosome protein concentrations ranging from 0.08 ng/mL to 2400 ng/mL were tested and we observed equal cell transformation for all concentrations with the exception of the two lowest, 0.08 ng/mL and 0.8 ng/mL. This indicates that *initiator* activity of cancer cell exosomes requires one dose of exosomes over a 3-day period with a protein concentration of at least 8 ng/mL (*Figure 2—figure supplement 1C*, *Figure 2—figure supplement 1—source data 3*). Furthermore, when cancer cell exosomes are boiled for 10 min at 100°C just prior to use as an *initiator*, the level of transformed foci decreases to background levels (*Figure 2—figure supplement 1A*, *Figure 2—figure supplement 1—source data 1*).

## Cancer-cell-derived exosomes function as a classic *initiator*

One common characteristic of *initiators* in the CTA is the capability of working with multiple *promoters* to induce cell transformation (*Fang et al., 2001*; *Sakai, 2007*). To test for this with the pancreatic cancer cell exosomes, we replaced TPA with another common *promoter*, cadmium chloride (CdCl$_2$) (*Fang et al., 2001*; *Keshava et al., 2000*; *Umeda et al., 1989*). We observed that NIH/3T3 cells treated with either MCA or Capan-2 exosomes as the *initiator* followed by treatment with CdCl$_2$ as the *promoter* resulted in formation of foci similar to that observed with TPA as the *promoter* (*Figure 2D*, *Figure 2—source data 2*). Treatment of cells with CdCl$_2$ alone resulted in background levels of foci, reiterating the fact that cell transformation is dependent on both *initiation* and *promotion*. These results indicate that exosomes isolated from pancreatic cancer cells act as general *initiators* in the transformation assay and are not dependent on a specific *promoter*.

## In vivo studies confirm the fully transformed state of cancer cell exosome-initiated cells

An important step of assessing the tumorigenic property of transformed cells is their ability to form tumors in vivo. To determine whether exosome-initiated transformed cells have the capacity to form tumors when injected subcutaneously into immunocompromised mice, we first isolated and expanded foci cells from the MCA/TPA and Capan-2 exosome/TPA experiments (*Figure 3A*). The cells from these foci were then injected into NSG (NOD scid gamma) mice at concentrations of $0.1 \times 10^6$, $0.5 \times 10^6$, and $2.5 \times 10^6$ cells to determine the sufficient cell density for tumor formation. Mice were followed for 37 days to measure tumor growth (*Figure 3—figure supplement 1A,B*). As a control, non-transformed NIH/3T3 cells were injected into mice at the highest concentration ($2.5 \times 10^6$ cells) (*Figure 3—figure supplement 1C*). After 37 days, tumor growth was observed in mice injected with cells from the chemically treated MCA/TPA foci and the cancer cell (Capan-2) exosome/TPA treated foci, at all three concentrations tested, whereas no tumor growth was observed in any of the mice injected with non-transformed control cells. In each case, the appearance of the tumors formed correlated with the number of cells injected, as higher concentrations of cells resulted in faster growth and larger tumors (*Figure 3—figure supplement 1A,B*). Histological analysis confirmed that the tumors are fibrosarcomas, as was expected because cells used in the CTA are NIH/3T3 cells from a mesenchymal origin (*Figure 3—figure supplement 1D*).

Additional in vivo studies were performed to analyze the tumor forming potential of a variety of foci, including background foci formed in control experiments (*Figure 3*, *Figure 3—figure supplement 1*, *Figure 3—figure supplement 2*). Using a concentration of $1 \times 10^6$ cells, we observed that cells collected from MIA PaCa-2 exosome/TPA, Panc-1 exosome/TPA, and BxPC-3 exosome/TPA foci formed tumors in mice at varying size and growth rates (*Figure 3D*). After injection of untreated background foci, we observed that 6 out of 15 total mice formed tumors. Notably, each of the six tumors (five from the same injected foci) grew later in the time course and at a significantly slower rate compared to *initiator and promoter* treated transformed foci cells, as has been previously observed (*Figure 3B*, *Figure 3—figure supplement 2A*) (*Xu and Rubin, 1990*).

## Proteomic analysis of initiated cells and transformed foci cells

We next used proteomics to analyze molecular changes in cells during the transformation process. Cells that were treated with MCA or Capan-2 exosomes as an *initiator* were harvested after the 3-day *initiator* treatment and total protein was analyzed by mass spectrometry. Untreated NIH/3T3 cells were used as a control. No marked global changes in protein content were observed for cells treated with MCA (yellow) or Capan-2 exosomes (blue) when compared to proteins found in untreated NIH/3T3 cells (*Figure 4—figure supplement 1*, *Figure 4—figure supplement 1—source data 1*).

Proteomics of transformed cells was also analyzed using mass spectrometry. Three foci from independent wells of both MCA/TPA transformed cells (yellow) and Capan-2 exosome/TPA transformed cells (blue) were compared to untreated NIH/3T3 cells as a control (*Figure 4*, *Figure 4—source data 1*). In total >1500 proteins were consistently identified in all three replicates of each sample type (see Materials and methods) (*Figure 4*). Proteins found in the transformed foci were compared to those found in the untreated control (*Figure 4A*). To determine whether overlapping proteins found in both control cells and foci are consistently present in each foci, we directly compared these

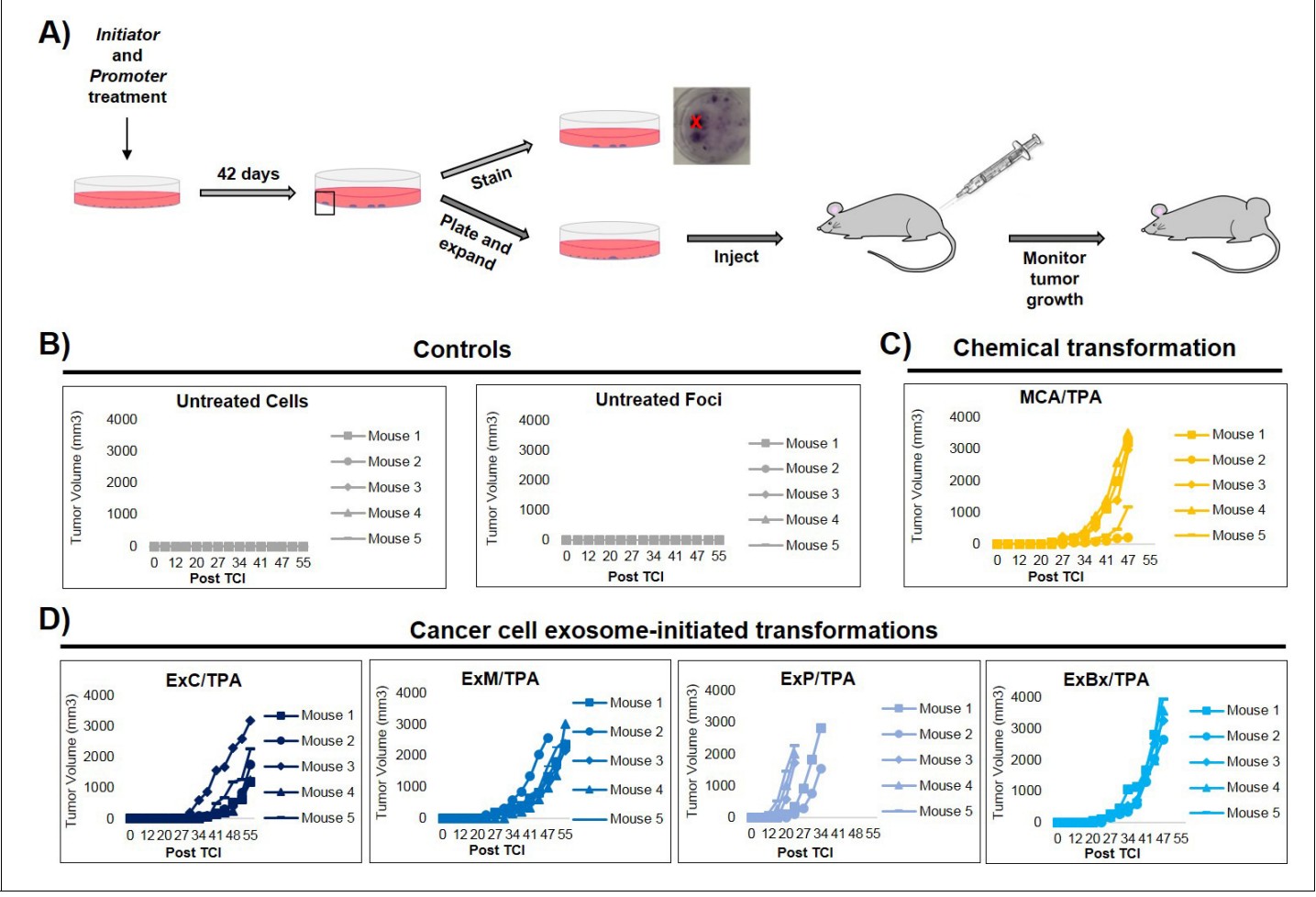

**Figure 3.** In vivo studies confirm the fully transformed form of cancer cell exosome-initiated cells. (**A**) In vivo assay. NIH/3T3 cells are treated with an initiator and promoter according to the cell transformation assay (42 days total). At the end of the transformation experiment, prior to methanol fixation and staining with crystal violet, foci were isolated, expanded, and established as a transformed cell line. Transformed cells are then subcutaneously injected into mice to monitor for tumor formation. Tumor growth was tracked by measuring tumor volume 2x/week for up to 55 days post injection (Post TCI) or until tumor size exceeded maximum limit. (**B**) Control mice include injection of untreated NIH/3T3 cells or background foci formed in untreated NIH/3T3 cells. Cells were injected at a concentration of $1 \times 10^6$ cells. Injection of untreated cells never resulted in tumor growth; injection of background foci from untreated cells resulted in tumor growth in 6 out of 15 total mice (see figure supplements for additional mice). (**C**) Results from injections of chemically transformed cells (MCA = initiator/TPA = promoter). Transformed cells were injected at a concentration of $1 \times 10^6$ cells. Tumor growth was observed in all mice (n = 5) from three independent experiments (see figure supplements for additional mice). (**D**) Results from injections of cancer cell exosome-initiated transformed cells; exosomes from four cancer cell lines, Capan-2 (ExC), MIA PaCa-2 (ExM), Panc-1 (ExP), and BxPC-3 (ExBx), were used as an initiator with the promoter TPA. Transformed cells were injected at a concentration of $1 \times 10^6$ cells. Tumor growth was observed in all mice (n = 5) for each treatment (see figure supplements for additional mice).

DOI: https://doi.org/10.7554/eLife.40226.014

The following figure supplements are available for figure 3:

**Figure supplement 1.** Additional in vivo studies of transformed cells.

DOI: https://doi.org/10.7554/eLife.40226.015

**Figure supplement 2.** Additional in vivo studies of transformed cells.

DOI: https://doi.org/10.7554/eLife.40226.016

data sets. The 'common' proteins found in both the untreated control and the transformed foci were compared using Venn diagrams and showed a high degree of overlap between foci of the same type (*Figure 4B*). By contrast, the proteins found to be 'unique' to each foci and absent in the control cells were compared and showed very little overlap between foci of the same type (*Figure 4C*). Gene ontology (GO) enrichment analysis was performed on the full set of proteins identified in each

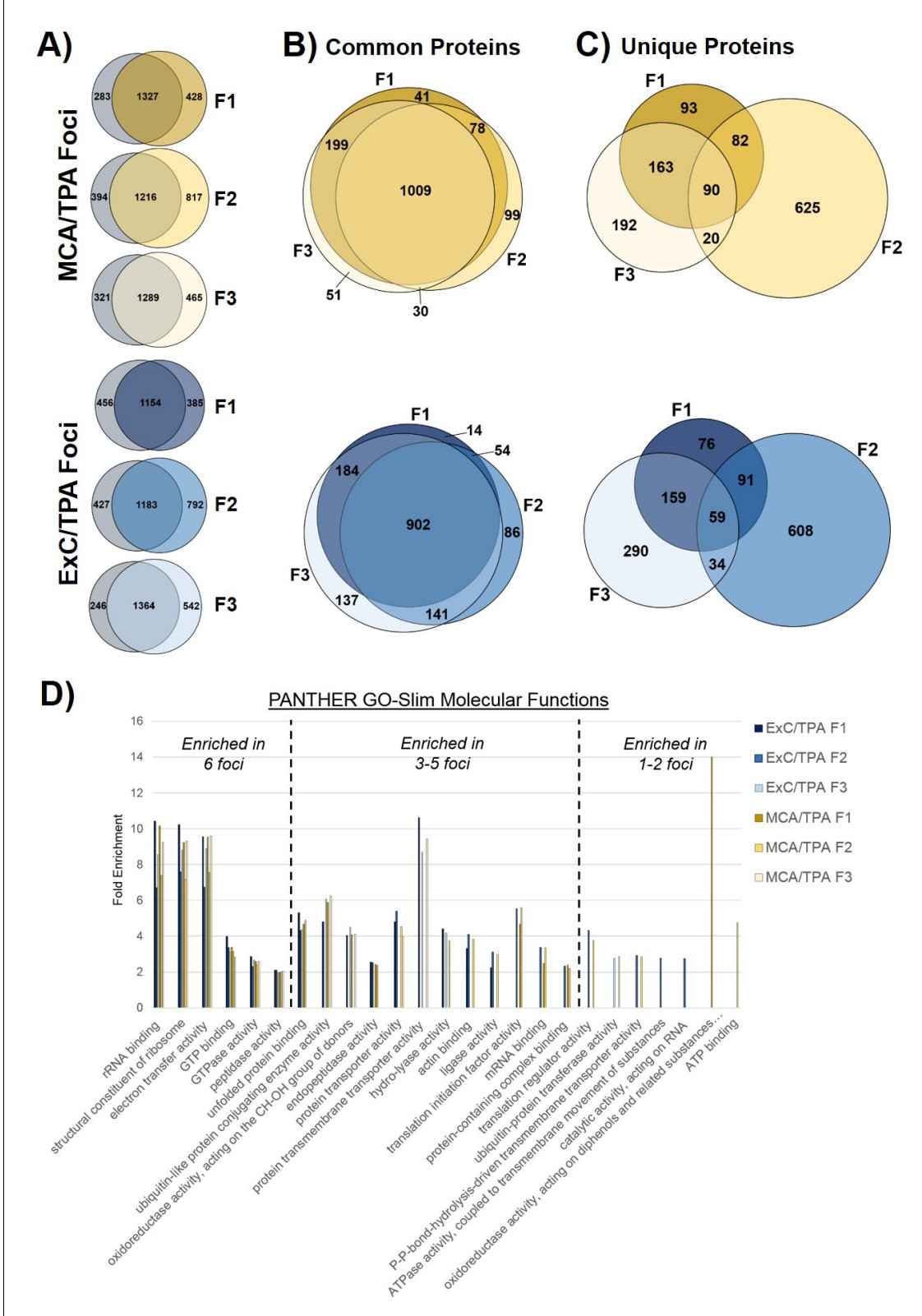

**Figure 4.** Proteomic profiling of transformed NIH/3T3 cells via mass spectrometry. (**A**) Comparison of proteins found in transformed cells resulting from treatment with both an initiator and promoter visualized by Venn diagrams. Three separate foci (F1, F2, F3) from MCA/TPA transformed cells and Capan-2 exosome (ExC)/TPA transformed cells were compared to untreated NIH/3T3 cells (control, gray). Results from three biological replicates were combined for each sample. (**B**) Comparison of common (overlap) proteins found in each of the six transformed foci samples; common proteins

*Figure 4 continued on next page*

*Figure 4 continued*

identified in control. (**C**) Comparison of unique proteins found in each of the six transformed foci samples; unique proteins are absent from control. (**D**) Gene Ontology enrichment analysis of proteins found in all six foci using PANTHER 14.0. Slim molecular functions identified as overrepresented based on analysis of proteins found in samples.

DOI: https://doi.org/10.7554/eLife.40226.017

The following source data and figure supplements are available for figure 4:

**Source data 1.** Relates to *Figure 4*.
DOI: https://doi.org/10.7554/eLife.40226.020
**Figure supplement 1.** Proteomic profiling of initiated NIH/3T3 cells via mass spectrometry.
DOI: https://doi.org/10.7554/eLife.40226.018
**Figure supplement 1—source data 1.** Relates to *Figure 4—figure supplement 1*.
DOI: https://doi.org/10.7554/eLife.40226.019

of the six foci in order to identify specific molecular functions overrepresented in the protein population (*Figure 4D*).

## Exome sequencing reveals mutagenic profiles

To better understand the genetic mechanism of cell transformation, exome sequencing was performed on the same set of transformed cells used for proteomic analysis and tumor mice studies: three MCA/TPA foci (yellow) and three Capan-2/TPA (blue) foci. In addition, three independent control foci were sequenced from untreated cells (gray) and TPA-only treated cells (gray); these are referred to as background foci. The total number of variants found by exome sequencing are visualized by Venn diagrams (*Figure 5—figure supplement 1*). The total set of variant data was used to generate the principle component analysis (PCA) plot shown in *Figure 5*. When all 12 foci samples were plotted together, the six transformed foci (MCA/TPA and Capan-2/TPA) appeared to cluster tightly while the six background foci (untreated and TPA-only) showed no clear relationship to one another (*Figure 5A*). When probed further, PCA of just the six transformed foci showed that there is no clear relationship between these samples (*Figure 5B*). Additionally, the total set of variant data was analyzed using MutaGene (*Goncearenco et al., 2017*) to investigate the specific types of nucleotide changes in the each of the 12 foci sequenced. MutaGene is a computational tool used to identify the most likely mutagenic processes associated with a set of variants found from whole exome or genome sequencing. The full mutational profile of each set of variants was decomposed into contributing COSMIC mutational signatures (*Figure 6*, *Figure 6—source data 1*). Clustering analysis of these signatures shows that all six transformed foci have similar mutational profiles that vary from the six background foci samples. The top mutational signatures found in each of the six transformed foci are COSMIC Signatures 20 and 15 (*Figure 6*), both of which are associated with defective DNA mismatch repair (MMR) and microsatellite instability. Foci from untreated cells and TPA-only treated samples did show signatures associated with microsatellite instability, but not as the top contributing signature. Instead, the top COSMIC Signature associated with each of the background foci was found to be COSMIC Signature 3, associated with failure of DNA double-strand break-repair. Considering that mismatch repair was found to be the top-contributing signature of each of the transformed foci, mismatch repair genes were analyzed in more detail for specific mutations (*Figure 6—figure supplement 1*). Missense mutations were found to be encoded in each of the transformed foci, but none of the foci contained the same mutations. None of the untreated background foci and only one of the TPA-only treated background foci had mutations in the analyzed MMR genes. Considering that mutations in oncogenes are often drivers of cell transformation, we also analyzed the mutational state of the 190 known oncogenes across the 12 foci samples. Results did not indicate a likely driver mutation as very few shared mutations were observed between independent foci of the same type and no common point mutations were found (*Figure 5—figure supplement 1*, *Figure 5—figure supplement 1—source data 1*).

## Discussion

There is a growing interest around exosomes functioning as information carrying cell messengers and as an active participant in cancer development and as a result, numerous studies have been

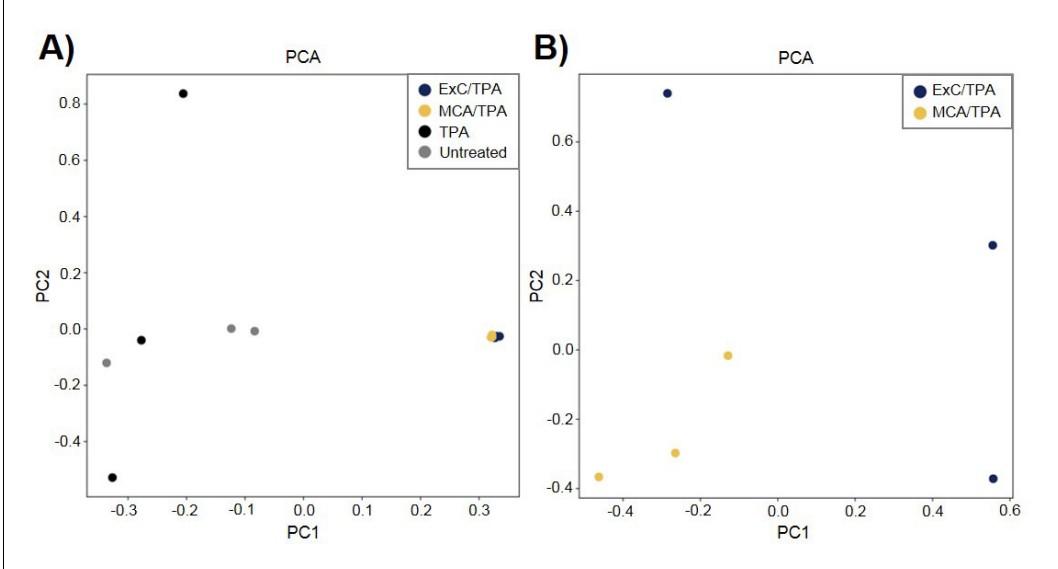

**Figure 5.** Principle component analysis (PCA) of transformed NIH/3T3 cells. (**A**) PCA plot showing relationship between three MCA/TPA transformed foci, three Capan-2 exosome (ExC)/TPA transformed foci, three control foci from TPA-only treated NIH/3T3 cells, and three control foci from untreated NIH/3T3 cells. (**B**) PCA plot showing relationship between same three MCA/TPA transformed foci and Capan-2 exosome (ExC)/TPA transformed foci in the absence of control samples. Principle component analysis is based on comparison of exome-seq variant data using PLINK's identity-by-state (IBS) estimates.

DOI: https://doi.org/10.7554/eLife.40226.021

The following source data and figure supplements are available for figure 5:

**Figure supplement 1.** Variants found by Exome-sequencing analysis.

DOI: https://doi.org/10.7554/eLife.40226.022

**Figure supplement 1—source data 1.** Relates to *Figure 5—figure supplement 1*.

DOI: https://doi.org/10.7554/eLife.40226.023

---

published describing their contribution to cancer progression (*Costa-Silva et al., 2015*; *Melo et al., 2014*; *Dai et al., 2018*; *Abdouh et al., 2017*; *Hamam et al., 2016*; *Antonyak et al., 2011*). In this study, we explored whether pancreatic cancer cell exosomes have a distinct role in the transformation of a normal cell to a malignant form by utilizing a classic two-stage cell transformation assay. We were able to describe a previously uncharacterized function for pancreatic cancer cell exosomes as an *initiator* in malignant cell transformation. Specifically, cancer cell exosomes were only shown to affect cell transformation when used as an *initiator* in combination with a *promoter*. Exosomes had no effect on cells when tested as a *promoter* or when used as both an *initiator* and *promoter* in the two-step CTA. Moreover, exosomes isolated from normal cells were unable to *initiate* cell transformation. By utilizing proteomics and exome sequencing, we were able to gain more understanding on the process of cell transformation. We propose that a single treatment of cancer cell exosomes on NIH/3T3 cells over 3 days can act in a similar way as the chemical *initiator* MCA, by randomly incorporating molecular changes into DNA. These random mutations then set the stage for a *promoter* to induce transformation of cells to form foci that are able to induce tumors when injected into mice.

For studies aiming to elucidate the biological functions of exosomes, it is essential that exosomes are reliably isolated from interfering cellular debris or contaminants. One of the current challenges in the exosome field is the lack of a universally accepted purification method. There are many published protocols on exosome purification from cells but it remains difficult to effectively isolate exosomes from contaminants (*Willms et al., 2016*; *Kowal et al., 2016*; *Théry et al., 2018*; *Vergauwen et al., 2017*). Major improvements have been made in recent years, one being the recent update to the Minimal Information for Studies of Extracellular Vesicles (MISEV2018) (*Théry et al., 2018*). In the current study, we aimed to meet the criteria posted in the MISEV2018 in

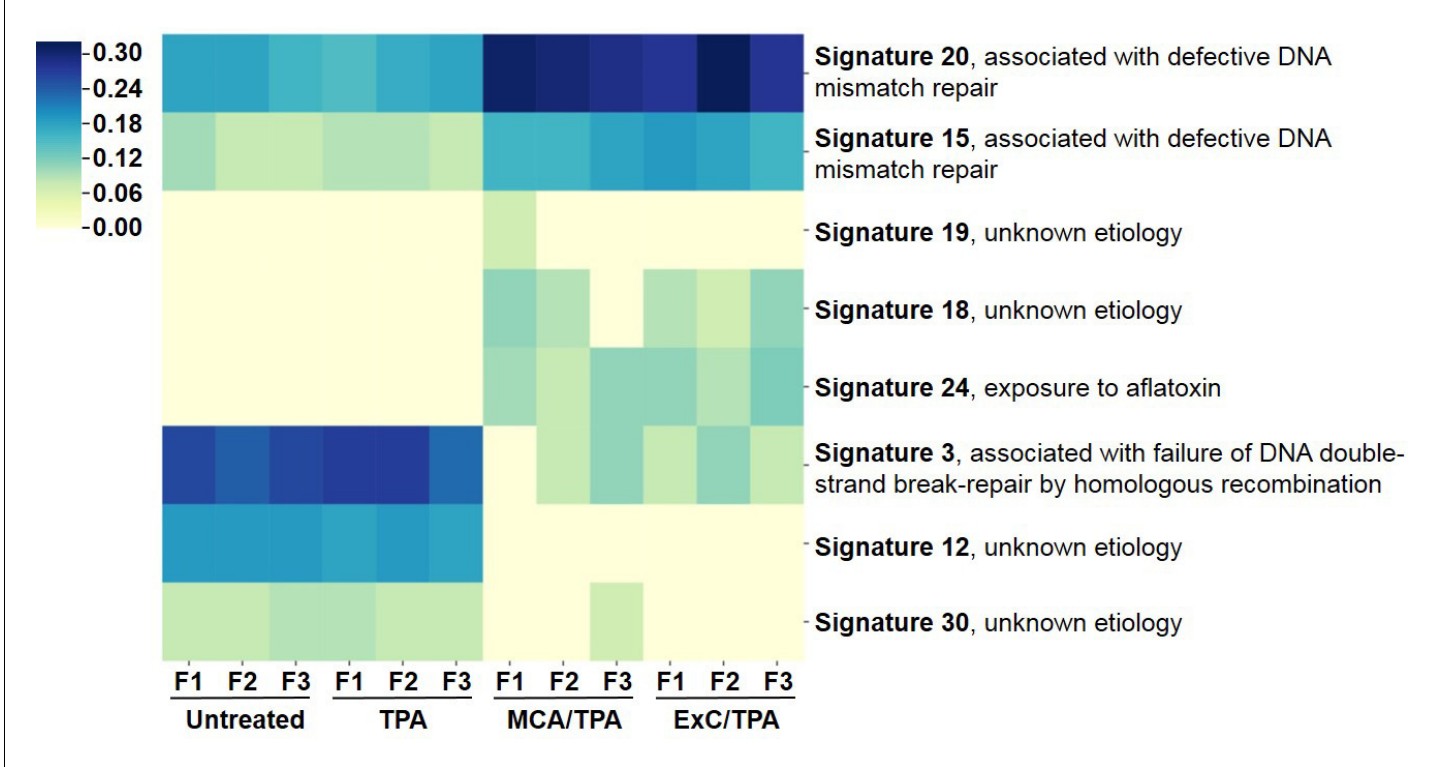

**Figure 6.** Mutational profiles of transformed NIH/3T3 cells. The 12 samples sequenced via Exome-seq include transformed foci formed from four treatment conditions on NIH/3T3 cells: untreated, TPA-only treated, MCA/TPA treated, and Capan-2 exosome (ExC)/TPA treated. The top COSMIC mutational signatures associated with each sample were identified using MutaGene and clustered based on similarity to generate the heatmap shown. Color range corresponds to the contribution score of each mutational profile.

DOI: https://doi.org/10.7554/eLife.40226.024

The following source data and figure supplements are available for figure 6:

**Source data 1.** Relates to *Figure 6*.
DOI: https://doi.org/10.7554/eLife.40226.027

**Figure supplement 1.** Non-synonymous variants found in mismatch repair associated genes (Msh2, Msh3, Msh6, Pms1, Pms2, Mlh1, Mlh3) across all 12 samples analyzed in *Figure 5*.
DOI: https://doi.org/10.7554/eLife.40226.025

**Figure supplement 2.** PROVEAN genome variant software was used to predict the potential impact of the identified missense variants on protein function in the mismatch repair associated genes.
DOI: https://doi.org/10.7554/eLife.40226.026

order to produce exosome preparations with minimal co-isolating contaminants. To eliminate the possibility that contaminants might participate in the exosome *initiator* activity, we further purified exosomes using a sucrose density gradient in order to obtain 'pure' exosome fractions (*Chiou and Ansel, 2016*). Identical experiments with both populations of exosomes demonstrated that both samples could act as an *initiator* in the CTA, thereby establishing exosomes as the *initiating* factor in this assay (*Figure 2—figure supplement 1B*).

The classic two-stage CTA was specifically used as a tool to study cell transformation because it is rigorous, reproducible, and has a well-established phenotype. This assay does not stimulate the entire in vivo neoplastic process, but it can provide essential information about identification of potential carcinogens and their mode of action. We also appreciate that the exosomes used in these assays are derived from human cancer cell lines whereas the 'normal' cells used in the CTA are NIH/3T3 cells from murine origin. Previous studies have shown highly conserved molecular functions across human and mouse, including the regulation of cell division, DNA replication, and DNA repair (*Monaco et al., 2015*). Through this assay, we were able to test the role that exosomes play in NIH/3T3 cell transformation in a well-controlled manner.

Recent studies demonstrate that cancer cell exosomes can in fact participate in cell transformation, however the details on how these exosomes contribute to transformation remained elusive. For example, Dai et al. showed that exosomes participated in the transformation process of normal cells after cells were treated with arsenite (*Dai et al., 2018*). Another study described a transfer of malignant traits to BRCA1 deficient human fibroblast when treated with either cancer patient sera or isolated cancer cell exosomes, leading to malignant transformation (*Abdouh et al., 2017*; *Hamam et al., 2016*). In addition, exosomes in breast cancer patient sera have been shown to promote nontumorigenic epithelial cells to form tumors in a Dicer-dependent manner (*Melo et al., 2014*). In each of the aforementioned studies, non-transformed cells were either pre-exposed to transforming agents or treated with cancer patient sera or medium from cultured cancer cells. Interestingly, Antonyak et al. has demonstrated that sustained treatment of NIH/3T3 cells with breast cancer or glioma cell-derived microvesicles has the ability to induce transformation of recipient cells (*Antonyak et al., 2011*). Contrastingly, in our studies, when cells were treated only with pancreatic cancer cell exosomes, increased cell transformation was not observed. While the specific details and effects on cell transformation attributed to exosomes vary between the studies, both demonstrate that exosomes are indeed playing a role in cell transformation.

Collectively, we observe that exosomes can function as an *initiator* in malignant cell transformation. Our results show that this activity is independent of the KRAS mutational state of exosome producing cells (*Figure 2C*). Boiling exosomes destroys the membrane structures and releases the vesicle content into the media, demonstrating that intact vesicles are needed to induce cell transformation (*Figure 2—figure supplement 1A*). Additionally, exosomes were found to work as a classic *initiator* and thus function with multiple *promoters* (TPA or $CdCl_2$) (*Figure 2D*). Finally, when cells isolated from MCA/TPA and cancer cell exosome/TPA foci were injected subcutaneously into immunocompromised mice, tumors formed (*Figure 3*, *Figure 3—figure supplements 1* and *2*).

We propose that, like MCA, cancer cell exosomes can cause random molecular changes to mediate the *initiator* step in cell transformation. Then, treatment with the *promoter* TPA forces cells to proliferate repeatedly and drives them to be fully transformed. The biochemistry associated with MCA-driven transformation is complex and diverse; it is known to produce bulky carcinogen-DNA adducts, which are associated with $G \cdot C \rightarrow T \cdot A$ transversions and thus introduces mutagenic and carcinogenic properties into DNA (*Malins et al., 2004*). According to Malins et al., MCA creates an identifiable tumor phenotype long before the appearance of tumors and causes changes in DNA structure that could be expected to influence gene expression and further translation (*Malins et al., 2004*). Although we observe no major changes in the protein content in *initiated* cells based on qualitative proteomic analysis, it is logical that changes could not be feasibly detected because they are occurring in specific proteins among specific cells in the total population. Additionally, it is known that exosomes do not uniformly contain the same pool of proteins, lipids, metabolites, or microRNAs, but rather each exosome contains a unique repertoire of biological molecules thus possibly causing a variety of changes in the cells (*Willms et al., 2016*; *Kowal et al., 2016*; *Smith et al., 2015*). Therefore, potentially both MCA and cancer cell exosomes are causing random molecular changes across the population of treated NIH/3T3 cells that cannot be detected amongst the overall protein composition of cells by mass spectrometry. Ultimately, more studies are needed to elucidate the molecular mechanism of this cancer cell exosome-mediated *initiation* event.

To further investigate the effects of cancer cell exosome-initiated transformation on cells, we probed the transformed foci that are formed during the two-stage cell transformation by proteomics and exome sequencing. Comparing the proteins found in MCA/TPA foci and Capan-2 exosome/TPA foci (transformed foci) revealed a set of unique proteins not identified in the control NIH/3T3 cells (*Figure 4A*). Venn diagrams comparing the proteins found in each foci reveal that these unique proteins vary between foci of the same type (*Figure 4B and C*). GO enrichment analysis of the transformed foci highlight this diversity, as a minority of molecular functions are found to be enriched in all the transformed foci while the majority of enriched functions vary between foci (*Figure 4D*). These studies, in contrast to the proteomic studies on the *initiator* treated cells, show major changes in the proteomic profiles of transformed cells (*Figure 4D*).

We performed exome sequencing analysis on twelve foci samples in total: three untreated control foci and three TPA-only treated foci (referred to as background foci); three MCA/TPA foci and three Capan-2 exosome/TPA foci (referred to as transformed foci). As expected, we observed diversity in mutational variants for each of these pools of cells because each foci was derived from independent

mutagenic events (*Figure 5—figure supplement 1*). PCA demonstrated that the transformed foci appear to cluster together while the background foci show no clear relationship to one another. Using PCA for just the transformed foci, we do observe heterogeneity between each sample, supporting the diversity in variants observed from sequencing (*Figure 5*, *Figure 5—figure supplement 1*). We then used MutaGene analysis with the full set of variants for each foci to generate the mutational profile for each. The data in *Figure 6* represents the relative contribution of the pan-cancer derived COSMIC signatures to the mutational profiles of each foci. Although all 12 samples are divergent at the molecular level, there is a distinct difference between the top contributing mutational signatures found in the six transformed foci as opposed to the six background foci (*Figure 6*).

The top signatures identified in the transformed foci (COSMIC Signatures 20 and 15), are associated with defective DNA mismatch repair and microsatellite instability (*Figure 6*). Consistent with this observation, analysis of MMR genes for specific mutations revealed mismatch repair gene mutations in these six transformed foci. Furthermore, based on in silico prediction analysis we observed that the mutations detected in mismatch repair genes may be deleterious or damaging for the activity of the mismatch repair proteins (*Figure 6—figure supplements 1* and *2*). Concurrently, sequencing data shows that the wild-type allele for each MMR gene is still present in these six foci. While MMR genes usually comply with Knudson's two-hit hypothesis for tumor suppressor genes, the presence of the wild-type copy of a MMR gene in somatic cells is not always sufficient for a normal function. Haploinsufficiency may be function-specific; for example, it has been demonstrated that responding to DNA damage requires a higher dosage of MMR protein than the repair function (*Peltomäki, 2016*). Regardless, the dysfunction of the MMR system is supported by the presence of the COSMIC profiles 20 and 15. Interestingly, none of the foci contain the same mutations in the MMR genes, further supporting the proposal that transformed cells originated from distinct mutagenic events. By contrast, only a single sample from the background foci was observed to have mutations (two) in MMR genes. In addition, the main driver found for all six background foci was COSMIC Signature 3. This profile associates with a failure in DNA double-strand break-repair that compromises genomic integrity and likely contributes to the high number of variants found in these samples (*Figure 5—figure supplement 1*).

Analysis of the top signatures for all six transformed foci shows that mutations in mismatch repair machinery are most likely what drives the transformation of these cells. Although the *initiators* (MCA and Capan-2 exosomes) varied for these six transformed foci, all were derived from NIH/3T3 cells and used TPA as a *promoter* for cell transformation. Analysis of the top signatures and perusal of mutated genes for the background foci demonstrates drivers other than MMR genes were used for the transformation of the cells. These observations support the proposal that when TPA is used as a *promoter (to drive proliferation)* in combination with a functional *initiator (an entity that creates a population of randomly mutated cells)*, the molecular path of cells towards tumorigenesis may not be random. Rather, TPA may drive a path in a subset of randomly mutated cells that accommodates a course towards COSMIC signatures 20 or 15. Furthermore, future studies might reveal that another *promoter* acting on *initiator* treated cells might lead cells towards another common endpoint with regards to the 30 different signature profiles.

A recent study by Felsentein et al. described an interesting finding in which a portion of co-occurring IPMNs in PDAC patients appeared to be genetically unrelated, meaning they shared no mutations in the assayed genes (*Felsenstein et al., 2018*). This elicits a fascinating question as to whether cells in a primary tumor could be derived from independent transformation events as opposed to exclusive clonal events (*Felsenstein et al., 2018*). Potentially, exosomes secreted by the primary tumor could orchestrate such events. Additional studies towards understanding the cancer cell exosome-mediated *initiation* are needed to address such questions, specifically, investigating the *initiation* capacity of cancer cell exosomes in more relevant cells like human epithelial cells.

In conclusion, we observe that cancer cell exosomes have the capacity to act as a classic *initiator* in the 2-stage CTA by incorporating random changes to NIH/3T3 cells over three days to mediate the first step in cell transformation (*Figure 7*). We observe that exosomes from pancreatic cancer cell lines, independently from the KRAS mutation status, can *initiate* the transformation of NIH/3T3 cells, while exosomes from normal pancreatic cells do not possess this ability. Future studies expanding on how cancer cell exosomes can uniquely function as an *initiator* are under way. Importantly, these observations provide insight into the molecular role of cancer cell exosomes in cell

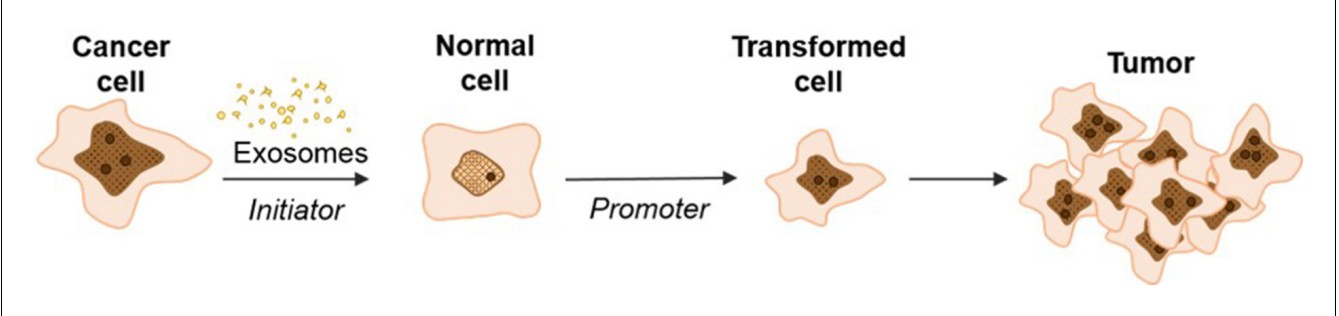

**Figure 7.** Schematic model of exosome mediated transformation. Exosomes secreted by cancer cells are taken up by normal NIH/3T3 cells and have the capacity to act as an *initiator* by incorporating random changes into the recipient cell genome. These *initiated* cells, when exposed to a *promoter*, can be induced by further alterations to a transformed state that has the ability to grow into a malignant tumor.
DOI: https://doi.org/10.7554/eLife.40226.028

transformation and how this activity might contribute to the dynamic conversation between normal and cancer cells.

# Materials and methods

## Key resources table

| Reagent type or resource | Designation | Source or reference | Identifiers | Additional information |
|---|---|---|---|---|
| Chemical | 12-*O*-Tetradecanoylphorbol-13-acetate (TPA) | Cell Signaling Technology | 4174 | |
| Chemical | Methylcholanthrene (MCA) | Sigma-Aldrich | 213942–100 MG | |
| Chemical | Cadmium Chloride (CdCl₂) | Sigma-Aldrich | 655198–5G | |
| Chemical | Dimethyl sulfoxide (DMSO) | Sigma-Aldrich | D2650—5 × 5 ML | |
| Cell line (*Mus musculus*) | NIH/3T3 | ATCC | RRID:CVCL_0594 | |
| Cell line (*Homo-sapiens*) | Dermal fibroblast (normal, Adult) | ATCC | PCS-201–012 | |
| Cell line (*Homo-sapiens*) | Capan-2 | ATCC | RRID:CVCL_0026 | |
| Cell line (*Homo-sapiens*) | PANC-1 | ATCC | RRID:CVCL_0480 | |
| Cell line (*Homo-sapiens*) | MIA PaCa-2 | ATCC | RRID:CVCL_0428 | |
| Cell line (*Homo-sapiens*) | BxPC-3 | ATCC | RRID:CVCL_0186 | |
| Cell line (*Homo-sapiens*) | HPDE (H6C7) | Kerafast | RRID:CVCL_0P38 | |
| Antibody | Anti-ALIX (3A9) (mouse monoclonal) | Abcam | Abcam, Cat#A2228, RRID:AB_10899268 | (1:500) |
| Antibody | α-actinin (H-2) (mouse monoclonal) | Santa Cruz Biotechnology | Santa Cruz Biotechnology, Cat#sc-17829, RRID:AB_626633 | (1:1000) |
| Antibody | Anti-β-actin (AC-74) (mouse monoclonal) | Sigma-Aldrich | Sigma-Aldrich, Cat#A2228, RRID:AB_476697 | (1:5000) |

*Continued on next page*

*Continued*

| Reagent type or resource | Designation | Source or reference | Identifiers | Additional information |
|---|---|---|---|---|
| Antibody | Calnexin (C5C9) (rabbit monoclonal) | Cell Signaling Technology | Cell Signaling Technology Cat# 2679, RRID:AB_2228381 | (1:1000) |
| Antibody | CD63 (rabbit polyclonal) | Proteintech | Proteintech, Cat#25682–1-AP, RRID:AB_2783831 | (1:1000) |
| Antibody | HSP90α/β (F8) (mouse monoclonal) | Santa Cruz Biotechnology | Santa Cruz Biotechnology, Cat#sc-13119, RRID:AB_675659 | (1:1000) |
| Antibody | TSG101 (4A10) (mouse monoclonal) | Thermo Fisher Scientific | Thermo Fisher Scientific Cat# MA1-23296, RRID:AB_2208088 | (1:500) |
| Software, algorithm | GraphPad Prism 8 | GraphPad | RRID:SCR_002798 | |
| Software, algorithm | Proteome Discoverer 2.1 | Thermo Scientific | Thermo Fisher Scientific, RRID:SCR_014477 | |
| Software, algorithm | FASTQC v0.11.5 | Babraham Bioinformatics | RRID:SCR_014583 | |
| Software, algorithm | Trim Galore | Babraham Bioinformatics | RRID:SCR_011847 | |
| Software, algorithm | Burrows-Wheeler Aligner (BWA) | Burrows-Wheeler Aligner | RRID:SCR_010910 | |
| Software, algorithm | Strelka2 | Illumina | RRID:SCR_005109 | |
| Software, algorithm | VCFtools | VCFtools | RRID:SCR_001235 | |
| Software, algorithm | SnpEff | SnpEff | RRID:SCR_005191 | |
| Software, algorithm | PLINK | *Purcell et al., 2007* | RRID:SCR_001757 | |
| Software, algorithm | Python | Python Software Foundation | RRID:SCR_008394 | |
| Software, algorithm | MutaGene | MutaGene | RRID:SCR_016574 | |
| Software, algorithm | PROVEAN | J.Craig Venter Institute | RRID:SCR_002182 | |

## Chemicals

12-$O$-Tetradecanoylphorbol-13-acetate (TPA) (Cell Signaling Technology), Methylcholanthrene (MCA) (Sigma-Aldrich), Cadmium Chloride ($CdCl_2$) (Sigma-Aldrich), Dimethyl sulfoxide (DMSO) (Sigma-Aldrich). Each compound was dissolved in DMSO and preserved at $-20°C$.

## Cells and culture conditions

Mouse embryo cell line, NIH/3T3, human primary dermal fibroblast, human pancreatic cancer cell lines: CAPAN-2, Panc-1, MIA PaCa-2, and BxPC-3, were purchased from American Type Culture Collection (ATCC, Manassas, VA). Immortalized human pancreatic duct epithelial cell line, HPDE, was from Kerafast (Kerafast, Boston, MA). Capan-2, MIA PaCa-2, and Panc-1 were maintained in Dulbecco's modified Eagle's medium (DMEM) (Millipore Sigma) supplemented with 10% (v/v) fetal bovine serum (FBS, Millipore Sigma) and 1% (v/v) antibiotics solution (Penicillin-Streptomycin, Millipore Sigma). BxPC-3 cells were maintained in minimum essential media (MEM) (Fisher) supplemented with 10% (v/v) fetal bovine serum (FBS) and 1% antibiotics solution (Penicillin-Streptomycin, Millipore Sigma). HPDE cells were maintained in Keratinocyte Serum-Free Media (KSFM, Invitrogen) with KSFM Supplements including epithermal growth factor (EGF) and bovine pituitary extract (BPE) (Invitrogen). Primary dermal fibroblast were maintained in Fibroblast Basal Medium (ATCC PCS-201–030) with Fibroblast Growth Kit-Serum-free (ATCC PCS-201–040) supplements included. NIH/3T3 cells were maintained in Dulbecco's modified Eagle's medium (DMEM) (Millipore Sigma) supplemented with 10% (v/v) Bovine Calf serum (CS, Gemini) and 1% antibiotics solution (Penicillin-Streptomycin, Millipore Sigma). All cell lines were cultured at 37°C in a humidified atmosphere of 5% $CO_2$.

Each cell line was tested free from mycoplasma. NIH/3T3 cells were used below passage 4 (p<4), primary dermal fibroblast (p<8), HPDE cells (p<8), and carcinoma cell lines (p<20).

## Experimental design

### Exosome isolation, subcellular fractionation, and TCA precipitation

#### Exosome isolation

Exosomes were isolated using a previously described combined ultrafiltration-ultracentrifugation protocol (*Adamczyk et al., 2011*). In detail, pancreatic cancer cells (CAPAN-2, Panc-1, MIA PaCa-2, and BxPC-3) and normal human cells (HPDE and human primary dermal fibroblasts) were grown in ten 225 cm$^3$ flasks in standard medium until they reached a confluency of approximately 70–80% (~3.5×10$^8$ cells). The carcinoma cell lines were then washed twice with medium and incubated in plain, serum-free medium for 72 hr. For HPDE cells and human primary dermal fibroblasts, phosphate-buffered saline (PBS) was used for washing and plain Keratinocyte SFM medium without supplements was used for exosome production for 72 hr. This protocol did not measurably increase the rate of cell death as determined by trypan blue exclusion, which showed over 93% live cell counts after 72 hr incubation in conditioned media. Next, the conditioned media (approximately 450 mL) from serum-free cell cultures were cooled down on ice, centrifuged (200x*g*, 10 min), and passed through 0.2 µm pore filters to remove cells, cell debris, and vesicles sized smaller than 220 nm. An inhibitor cocktail was added to protect the proteins from proteolytic digestion (PMSF and inhibitor cocktail complete Roche, Mannheim, Germany). Enrichment of exosomes was accomplished by subsequent ultrafiltration with Amicon Ultra 100K (4000x*g*, 25 min, 4°C), followed by ultracentrifugation at 120,000 g for 90 min at 4°C. The exosome pellet was washed in PBS followed by another ultracentrifugation at 120,000 g for 90 min at 4°C. The final exosome pellet was resuspended in PBS and validated by characterization via mass spectrometry analysis of proteins, western blot analysis, electron microscopy analysis (TEM), and nanoparticle tracking analysis (NTA) (*Lötvall et al., 2014*; *Witwer et al., 2017*; *Théry et al., 2018*). Each exosome pellet, after final resuspension into PBS, was divided into 5 µl aliquots, stored at −80°C, and thawed right before use. The protein concentration of the exosome fraction was measured after each exosome isolation using CBQCA protein quantitation kit (Invitrogen). Exosomes were used in the cell transformation assay experiments at a concentration of 80 ng/mL (equivalent to $7 \times 10^7$ particles/mL) based on dose response studies.

#### Sucrose gradient separation

In addition to the exosome isolation protocol described above, exosomes were further purified using floatation into a sucrose density gradient (*Chiou and Ansel, 2016*). In detail, sucrose gradients were built manually as described in reference five by first preparing 12 sucrose stock fractions in PBS with sucrose concentrations ranging from 10–90%. Half of the exosome pellet from the isolation protocol described above was resuspended in 50 µl PBS with 1 ml of 90% sucrose stock solution and loaded at the bottom of a 13.2 mL ultra-clear Beckman ultracentrifuge tube. The gradient was layered by sequentially pouring 1 mL of each of the remaining 11 solutions in order from highest to lowest sucrose concentration. Tubes were centrifuged for 16 hr at 4°C at 100,000 g (24200 rpm) in a TH-641 rotor. At completion, six 2 mL fractions were collected from each of the tubes. Next, 9 mL of PBS was added to each of the 2 mL fractions and centrifuged at 4°C at 100,000 g for 1 hr. The supernatant was carefully aspirated before the pellet was resuspended in 50 µl of PBS and validated by mass spectrometry, western blot, and nanoparticle tracking analysis (NTA). The protein concentration of the fractions was measured using CBQCA protein quantitation kit (Invitrogen). Fraction 3, shown to contain exosomal marker proteins, was used in the cell transformation assay experiment at 80 ng/mL.

#### Subcellular fractionation

Cells from exosome preparations were harvested from one 225 cm$^3$ flask, after the conditioned media for exosome isolation was collected, using 0.25% trypsin-EDTA (Sigma-Aldrich) treatment, followed by lysing according to previously published protocol (*Casey et al., 2018*). In brief, cells were suspended in HNMEK lysis buffer (20 mM HEPES pH 7.4, 50 mM NaCl, 2 mM MgCl$_2$, 2 mM EDTA, 10 mM KCl, 50 nM EGTA, protease inhibitors) and lysed using a Dounce homogenizer. Lysates were centrifuged at 500 g for 10 min at 4°C to remove nuclei and cellular debris. The supernatant was

collected and centrifuged at 10,000 g for 10 min to pellet ER and mitochondrial membranes (P2 fraction). Cytoplasmic S2 fraction was collected and kept for further analysis. P2 fraction was washed once in HNMEK lysis buffer, centrifuged at 10,000 g for 10 min, and the pellet was resuspended in RIPA lysis buffer (50 mM Tris pH 8, 150 mM NaCl, 5 mM EDTA pH8, 1% Nonidet P-40, 0.5% deoxycholate, 0.1% SDS, 1 mM PMSF, protease inhibitors). Protein concentration was measured from each fraction using the Bradford Protein assay (Bio-Rad).

## TCA precipitation

The last fraction collected from the exosome isolation procedure was the flow through media (M). This flow through consists of 20% starting conditioned media after the ultrafiltration step, and was kept for deoxycholate-trichloroacetic acid precipitation (DOC-TCA) of the proteins. This medium represents a sample that should not contain any exosomal proteins when analyzed for exosome markers using western blot analysis. In brief, 0.15% DOC was added to media samples (1:10 ratio) and incubated on ice for 15 min followed by addition of TCA to 8% final concentration and incubated overnight at 4°C. Precipitated proteins were pelleted by centrifuging at 18,000 rpm and washed twice with cold acetone, prior to re-suspension in 10 mM Tris–HCl pH = 8.0. The protein concentration was measured using CBQCA protein quantitation kit (Invitrogen).

## Cell transformation assay

The two-stage cell transformation assay was carried out according to the protocol described by *Sakai and Sato (1989)* with small modifications. In detail, the frozen stock of NIH/3T3 cells were thawed and cultured. Actively growing cells with passage number <4 were seeded for the transformation assay at a density of $2.5 \times 10^3$ cells/well in a 6-well plate with 2 mL of culture medium. Two days after seeding, media was replaced with media containing an *initiator*, either MCA (0.5 μg/mL), exosomes (80 ng/mL), or 0.02% DMSO, and cells were grown for 3 days. Next, the medium was replaced with fresh medium and cells were grown for an additional 2 days. Cultures were then treated with a medium containing a *promoter*, either TPA (300 ng/mL), exosomes (80 ng/mL), CdCl$_2$ (120 ng/mL), or 0.2% DMSO, for 2 weeks. The cells were subsequently cultured in normal medium for 3 weeks. The medium was changed every other day during the *promoter* treatments and twice a week for the last 3 weeks of the experiment. The cells were fixed with methanol and stained with crystal violet for focus scoring. Each test chemical was dissolved in DMSO. The concentration of vehicle was below 0.2%, which did not affect the induction of transformed foci.

## Foci scoring

The scoring of transformed foci was carried out according to established criteria on focus scoring (*Sasaki et al., 2012*). Different categories of foci that can be observed are Type I, Type II, and Type III. Only Type III foci are scored as malignantly transformed and were counted as positive in this study. Foci were assessed for the following characteristics: deep basophilic staining, spindle-shaped cells, multilayer growth (piling up of cells), random orientation at the edge of the focus, and invasive growth into the background monolayer; each characteristic needed to be present for Type III classification. To ensure accurate scoring, foci scoring was performed in a double-blinded manner by two researchers.

## Tumorigenicity of transformed cells

All animals were housed in a pathogen-free facility with 24 hr access to food and water. Experiments were approved by, and conducted in accordance with, an IACUC approved protocol at UT Southwestern. Six-to-8-week-old female NOD/SCID mice were obtained from an on-campus supplier.

At the end of the transformation experiment, prior to methanol fixation and staining with crystal violet, MCA/TPA-treated and cancer cell exosome/TPA-treated cells that had formed type III foci were isolated, expanded, and established as a transformed cell line. To assess their tumorigenic property, cells were injected subcutaneously into mice. To determine the sufficient cell density for tumor formation, cells were first isolated from MCA/TPA-treated and Capan-2 exosome (ExC)/TPA-treated experiments and injected into mice at three different cell concentrations ($0.1 \times 10^6$, $0.5 \times 10^6$, and $2.5 \times 10^6$ cells). Untreated NIH/3T3 cells were injected into mice as a control at the highest concentration used ($2.5 \times 10^6$ cells). Every group consisted of n = 5 mice. Mice were observed for

tumor formation by palpating twice a week and recording the weight of the mice and size of the tumor. The experiment was terminated 37 days post injection, when the tumor size in some of the animals had reached maximum allowable diameter. Histological analyses confirmed that the tumors are fibrosarcomas, as expected from the transformed cells of mesenchymal origin. Additional in vivo experiments include subcutaneous injection of mice (n = 5/group) with $1 \times 10^6$ cells isolated after treatment with MIA PaCa-2 (ExM)/TPA, Panc-1 (ExP)/TPA, or BxPC-3 (ExBx)/TPA as well as injection of three background foci formed from untreated NIH/3T3 cells. Tumor growth was tracked in the same manner up to 55 days post injection or until tumor growth exceeded maximum allowable size.

### Initiation assay

NIH/3T3 cells were plated on a six-well plate at a density of $2.5 \times 10^3$ cells/well with 2 mL of culture medium. Two days after seeding, media was replaced with fresh complete media or media containing an *initiator*; either 0.5 µg/mL MCA or 80 ng/mL Capan-2 exosomes, and cells were grown for 3 days, followed by 2 days of recovery (two wells for each condition). Next, cells were harvested and lysed with RIPA lysis buffer (50 mM Tris pH 8, 150 mM NaCl, 5 mM EDTA pH 8, 1% Nonidet P-40, 0.5% deoxycholate, 0.1% SDS, 1 mM PMSF, protease inhibitors) by incubating cells on ice for 30 min and vortexing twice during the incubation. At the end of incubation, lysates were centrifuged at 10,000 g to pellet cell debris. Protein concentration was measured using the Bradford Protein assay (Bio-Rad). Experiments were performed in triplicate for each treatment condition. Total protein composition (26 µg of protein/sample) was analyzed by mass spectrometry.

### Preparation of samples for mass spectrometry and exome sequencing analysis
#### Proteins

For analysis of exosomal proteins by mass spectrometry, exosome samples (equivalent of approximately 15 µg of protein) isolated from each cell line were thawed from −80˚C storage and 10 µl of $5 \times$ protein sample buffer was added. Samples were boiled for 5 min and loaded on TGX stain-free gels (Bio-Rad) and run 10 mm into the top of an SDS–PAGE gel. Gel bands containing proteins were excised for mass spectrometry analysis.

For analysis of the total protein composition of untreated NIH/3T3 cells, transformed cells, and cells from the initiation assay, cells were lysed in RIPA lysis buffer and protein concentration was measured using the Bradford Protein assay (Bio-Rad). Equal amounts of proteins (approximately 26 µg of protein from the initiation assay samples and 35 µg of protein from the transformed cell samples) from three biological replicates were taken and $5 \times$ protein sample buffer was added. Samples were boiled for 5 min and loaded on TGX stain-free gels (Bio-Rad) and run 10 mm into the top of an SDS–PAGE gel. Gel bands containing proteins were excised for mass spectrometry analysis.

#### Genomic DNA

DNA for exome sequencing was extracted from untreated NIH/3T3 cells, three background foci from untreated NIH/3T3 cells, three background foci from TPA-only treated NIH/3T3 cells, three foci from MCA/TPA treated NIH/3T3 cells, and three foci from ExC/TPA treated NIH/3T3 cells using Quick-gDNA MiniPrep Kit (Genesee Scientific) according to the manufacturer's protocol. Quality and concentration of the DNA was measured with BioAnalyzer. Exome libraries were prepared using IDT xGen Exome Research Panels and sequenced on Illumina HiSeq 4000 at 100x coverage.

## Data analysis
### Validation of exosome isolation
#### Transmission electron microscopy (TEM)

Electron microscopy was used to characterize vesicles pelleted by the ultrafiltration-ultracentrifugation isolation method described above and to provide information on the size of the vesicles. TEM negative staining was performed on the aliquots stored at −80˚C. 10 µl of exosome suspension in PBS was placed onto carbon-coated grids (200mesh) for 1 min and negatively stained with 2% uranyl acetate solution for 1 min. Grids were visualized at 13000x to 68000x in a FEI Tecnai G$^2$ Spirit transmission electron microscope at 120kV. Separate images were taken to provide a wide field encompassing multiple vesicles or to provide close-up images of single vesicles.

### Nanoparticle tracking analysis

Nanoparticle tracking analysis (NTA) was performed using NanoSight Version 2.3 on crude exosomes and exosomes further purified by sucrose density gradient (Fraction three only). Finite Track Length Analysis (FTLA) was used for size determination. In each case, average vesicle size per concentration was determined from three measurements of a single exosome preparation.

### Mass spectrometry

Mass spectrometry analysis was performed on crude exosomes derived from Capan-2 cells or purified exosomes (Fraction 3) derived from Capan-2 cells. *Figure 1—figure supplement 2* shows a list of the top 20 most commonly found proteins in exosomes according to the Exocarta database (*Kowal et al., 2016*). All twenty proteins were identified by mass spectrometry in both populations of Capan-2 cell exosomes. Complete proteins lists from crude exosomes and Fraction three exosomes are included as *Figure 1—figure supplement 2—source datas 1* and *2*.

### Antibodies and western blot

Alix antibody (cat. no. ab117600, Abcam), α-actinin antibody (H-2, cat. no. sc-17829, Santa-Cruz), β-actin antibody (Clone AC-74, cat. no. A2228, Sigma-Aldrich), Calnexin antibody (Clone C5C9, cat. no. 2679, Cell Signaling), CD63 antibody (cat. no. 25682–1-AP, Proteintech), HSP90α/β (F-8, cat. no. sc-13119, Santa-Cruz), TSG101 antibody (Clone 4A10, cat. no. MA1-23296, Thermo Fisher).

Western blots were used to examine the presence of common exosomal proteins in cellular fractions and sucrose gradient purified exosomes. Using Capan-2 cells as a representative example, equivalent micrograms of proteins from ER and mitochondrial (P2), cytoplasmic (S2), media (M), and exosome (Ex) fractions, prepared from different steps during exosome isolation as described above in section 1.1, were separated by SDS-PAGE and transferred to nitrocellulose membranes. Development was performed using Pierce ECL 2 Western blotting substrate (Thermo Fisher Scientific) and radiographic films (Lightlab).

### Cytotoxicity and colony formation efficiency of cadmium chloride (CdCl$_2$)

Cytotoxicity and colony formation efficiency assays were used to determine a suitable concentration of CdCl$_2$ for use as a *promoter* in cell transformation assay. Cytotoxicity of CdCl$_2$ was determined using a protocol as described by *Umeda et al. (1989)* and *Fang et al. (2001)* with some modifications. Experiments were repeated twice to confirm reproducibility. In brief, toxicity was tested by plating $2 \times 10^4$ cells/mL into 6-wells and culturing for 24 hr. Next, cells were treated with 40, 120, 240, or 360 ng/mL CdCl$_2$, with 3-wells/each concentration. After a 4 day cultivation, the cell number of each well was determined after trypsin treatment using trypan blue exclusion method by counting the number of live cells to the number of dead cells. Cell viability remained high (>89%) for the first three concentrations (40, 120, 240 ng/mL) but dropped to an average of 56% in the highest concentration (360 ng/mL).

For the colony formation efficiency assay, cells were plated at 200 cells/2 mL into 6-wells and cultured for 24 hr. Next, cells were treated with 40, 120, 240, or 360 ng/mL CdCl$_2$, 3-wells/each concentration, for 10 days. Media was changed on day 5. The cells were fixed with methanol and stained with crystal violet for counting the number of colonies. Only colonies comprising >50 cells were scored. Relative colony formation efficiency was calculated as (%) = (number of test colonies/number of control colonies) x 100 (*Fang et al., 2001*). At the two highest concentrations (240, 360 ng/mL) colony formation was reduced to 4% when compared to control, untreated cells. At 120 ng/mL colony formation was inhibited by 50% and cell viability maintained above 89%. The lowest concentration, 40 ng/mL, did not have any effect on the colony formation efficiency when compared to control colony formation. The highest CdCl$_2$ concentration to inhibit colony formation by 50% while retaining cell viability was 120 ng/mL and was therefore used in cell transformation assays.

### Dose response studies

Dose response experiments were performed to determine a suitable concentration of exosomes for use as an *initiator* in cell transformation assays. Protein concentration was used as a normalization strategy. Transformation assays with different protein concentrations of exosomes were repeated twice to confirm reproducibility. Cells were treated with 0.08, 0.8, 8, 24, 80, 240, 800 or 2400 ng/mL

of exosomes as an *initiator* as described in detail in section 1.2; 6-wells per concentration were used. At the end of the cell transformation assay, cells were fixed with methanol and stained with crystal violet. Focus scoring were performed according to the method described in section 1.3. Equal transformation induction activity was observed for all concentrations except the two lowest ones, 0.08 ng/mL and 0.8 ng/mL. A standard concentration of 80 ng/mL (equivalent to $7 \times 10^7$ particles/mL) of proteins were used in each transformation assay.

## Statistical analysis of foci scoring
Statistical analysis of foci scoring was performed by two-tailed unpaired t test with Welch's correction.

## Proteomic analysis by mass spectrometry
Sample preparation for mass spectrometry analysis included the excision of proteins from polyacrylamide gels via SDS-PAGE and Coomassie blue dye staining. Protein samples were reduced and alkylated using DTT and iodoacetamide, respectively. Samples were digested overnight using trypsin (37°C) and resulting peptides were de-salted using solid phase extraction (SPE). LC-MS/MS experiments were performed on a Thermo Scientific EASY-nLC 1200 liquid chromatography system coupled to a Thermo Scientific Orbitrap Fusion Lumos mass spectrometer. To generate MS/MS spectra, MS1 spectra were first acquired in the Orbitrap mass analyzer (resolution 120,000). Peptide precursor ions were then isolated and fragmented using high-energy collision-induced dissociation (HCD). The resulting MS/MS fragmentation spectra were acquired in the ion trap. MS/MS spectral data from exosome samples was searched using Proteome Discoverer 2.1 software (Thermo Scientific) against entries included in either the Human Uniprot protein database (173,060 entries) (exosome samples) or the Mus musculus (Mouse) Uniprot protein database (86,520 entries) (NIH/3T3 cell samples). Search parameters included Carbamidomethylation of cysteine residues (+57.021 Da) as a static modification and oxidation of methionine (+15.995 Da) and acetylation of peptide N-termini (+42.011 Da) as dynamic modifications. The precursor ion mass tolerance was set to 10 ppm and the product ion mass tolerance was set to 0.6 Da for all searches. Peptide spectral matches were adjusted to a 1% false discovery rate (FDR) and proteins were filtered to a 1% FDR.

For exosome samples, only a single run was analyzed for the presence of exosomal marker proteins. For transformed NIH/3T3 cells (*Figure 4*) and initiated NIH/3T3 cells (*Figure 4—figure supplement 1*) each sample was run in biological triplicate. Additional filtering was applied to protein datasets from transformed NIH/3T3 cells (data contained in *Figure 4*) to only compare proteins identified in all three biological replicates. Complete proteins lists from crude exosomes and Fraction three exosomes are included as *Figure 1—figure supplement 2—source datas 1* and *2*. Complete protein lists from initiated NIH/3T3 cells are included as *Figure 4—figure supplement 1—source data 1*. Complete protein lists from transformed NIH/3T3 cells are included as *Figure 4—source data 1*.

Gene ontology (GO) enrichment analysis was performed using PANTHER 14.0 on the set of proteins identified in each of the six transformed foci shown in *Figure 4* (three MCA/TPA foci and three ExC/TPA foci). Each protein list contained proteins identified in all three biological replicates for each foci. Protein lists were searched against the reference *Mus musculus* protein database to identify overrepresented molecular functions (*Ashburner et al., 2000*; *The Gene Ontology Consortium, 2019*; *Mi et al., 2017*).

## Exome sequencing analysis
### Variant calling
Raw reads were quality controlled and filtered using FASTQC v0.11.5 (*Andrews, 2016*) and Trim Galore v0.4.1 (*Krueger, 2015*) using default settings. Reads were mapped to the mouse reference genome (GRCm38) using BWA-MEM v0.7.12 (*Li and Durbin, 2009*). Somatic mutations were called using Strelka2 v2.9.0 against the original NIH/3T3 cells (*Kim, 2018*). Mutations were filtered for quality using VCFtools v0.1.14 that meet PASS criteria based on Empirical Variant Score and minimum read depth of DP >10. Comparison between similarly treated foci were compared using 'vcf-compare' using VCFtools v0.1.14 and Venn diagrams comparing all variants were drawn using Python v3.6.8.

## Annotation and foci comparison

Mutations were annotated using SnpEff v4.3q (PMID:22728672) for loss-of-function or missense mutation. Mutation-based clustering analysis was performed on full variant data set for each foci using pairwise identity-by-state function in PLINK v.1.90b4 (*Purcell et al., 2007*) and visualized using the first two principal components using Python v3.6.8 to compare similarity between foci. Annotations were filtered for loss-of-function or missense mutation in order to generate the table of non-synonymous variants found in mismatch repair associated genes in *Figure 6—figure supplement 1*.

## Generating mutational signatures and heatmap

Mutational signatures were derived from the full set of variants for individual samples using Muta-Gene (*Goncearenco et al., 2017*). To identify overlapping signatures between samples, we performed hierarchical clustering by calculating the Euclidean distance using clustermap from seaborn v0.7.1[@doi:10.5281/zenodo.54844].

## In silico analysis

PROVEAN v1.1.3. (Protein Variation Effect Analyze, http://provean.jcvi.org/index.php) genome variant software was used to predict the potential impact of the identified missense variants on protein function in the mismatch repair associated genes. This tool provides PROVEAN and SIFT predictions for a list of genome variants.

## Data and materials availability

All data are available in the main text or the supplementary materials.

## Acknowledgements

We thank members of the Orth lab for their helpful discussions and advice. We thank John Minna for his critical assessments and support. We thank Diego Castrillon for his expert advice on tissue pathology and the UTSW Electron microscopy Core Facility. In memory and thanks to our friend and colleague Dr. Alfred G Gilman.

## Additional information

### Competing interests

Kim Orth: Reviewing editor, *eLife*. The other authors declare that no competing interests exist.

### Funding

| Funder | Grant reference number | Author |
| --- | --- | --- |
| National Institutes of Health | GM115188 | Kim Orth |
| Once Upon A Time Foundation | | Kim Orth |
| Welch Foundation | I-1561 | Kim Orth |
| National Institutes of Health | CA192381 | Rolf Brekken |

The funders had no role in study design, data collection and interpretation, or the decision to submit the work for publication.

### Author contributions

Karoliina Stefanius, Conceptualization, Resources, Data curation, Formal analysis, Validation, Investigation, Methodology, Writing—original draft, Writing—review and editing; Kelly Servage, Conceptualization, Data curation, Software, Formal analysis, Validation, Investigation, Methodology, Writing—original draft, Writing—review and editing; Marcela de Souza Santos, Data curation, Validation, Investigation, Methodology, Writing—original draft, Writing—review and editing; Hillery Fields Gray, Data curation, Formal analysis, Validation, Investigation, Methodology, Writing—original draft,

Writing—review and editing; Jason E Toombs, Suneeta Chimalapati, Data curation, Investigation, Methodology; Min S Kim, Data curation, Software, Validation, Investigation, Methodology, Writing—review and editing; Venkat S Malladi, Data curation, Formal analysis, Methodology, Writing—review and editing; Rolf Brekken, Resources, Data curation, Formal analysis, Supervision, Funding acquisition, Validation, Investigation, Methodology, Writing—review and editing; Kim Orth, Conceptualization, Formal analysis, Supervision, Funding acquisition, Investigation, Writing—original draft, Writing—review and editing

## Author ORCIDs

Karoliina Stefanius https://orcid.org/0000-0002-9456-1954
Kelly Servage https://orcid.org/0000-0001-7183-2865
Rolf Brekken http://orcid.org/0000-0003-2704-2377
Kim Orth https://orcid.org/0000-0002-0678-7620

## Ethics

Animal experimentation: This study was performed in accordance with the recommendations in the Guide for the Care and Use of Laboratory Animals of the National Institutes of Health and according an approved UT Southwestern institutional animal care and use committee (IACUC) protocol (APN 2016-101732).

## Decision letter and Author response

Decision letter https://doi.org/10.7554/eLife.40226.031
Author response https://doi.org/10.7554/eLife.40226.032

## Additional files

### Supplementary files
• Transparent reporting form
DOI: https://doi.org/10.7554/eLife.40226.029

### Data availability
All data generated or analysed during this study are included in the manuscript and supporting files.

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
