## [Decision Letter]

[**Editorial note:** This article has been through an editorial process in which the authors decide how to respond to the issues raised during peer review. The Reviewing Editor's assessment is that major issues remain unresolved.]

Decision letter after peer review:

Thank you for submitting your article "Cancer cell exosomes can initiate malignant cell transformation" for consideration by *eLife*. Your article has been reviewed by three peer reviewers, one of whom is a member of our Board of Reviewing Editors, and the evaluation has been overseen by Jeffrey Settleman as the Senior Editor, with additional comment by Randy Schekman (Editor-in-Chief). The reviewers have opted to remain anonymous.

The Reviewing Editor has highlighted the concerns that require revision and/or responses, and we have included the separate reviews below for your consideration. If you have any questions, please do not hesitate to contact us.

Summary:

In this manuscript, the authors provide evidence that exosomes derived from pancreatic cancer cells can act as initiators of tumorigenesis. The primary evidence for this comes from a transformation assay in NIH 3T3 cells. This work adds to the growing body of evidence that exosomes can make major contributions to overall tumorigenic phenotypes.

Major concerns:

There were several major and significant concerns raised by the reviewers. They are outlined in detail below, but in general there were serious reservations about the informatic analysis and how it was performed, and a concern as to whether the mismatch repair interpretation is robust. More importantly, even if the informatics is corrected, there is no mechanistic insight into how these defects in mismatch repair occur, or even the necessity/sufficiency of this pathway.

After the review was completed, the Editor-in-Chief expressed additional concerns about the conclusions as drawn on the basis of experiments with a crude fraction of particulate material from conditioned medium:

Your claims are based on the use of "exosomes" as defined by what you refer to as established protocols for their isolation. Unfortunately, the standards in the field are inadequate and fail to account for many decades of experience in the fractionation of membrane organelles (please see Shurtleff et al., Annual Reviews of Cancer Biology, 2018). The crude particulate fraction obtained by filtration and differential centrifugation of conditioned medium contains other material including contaminating membranes and nonmembranous particles such as large RNPs. Thus, without further fractionation of the relevant sedimentable species, whether membranous or not, your conclusions about the roe of exosomes in the initiation of tumorigenesis will remain suspect.

As you are aware, in this *eLife* trial, we are committed to eventually publishing the paper if you are able to address these issues fully. However, given the extent of what the reviewers are requesting to make the manuscript suitable for *eLife*, you may want to consider withdrawing this work for further investigation. This is your decision, but please let us know your thoughts in either case.

Separate reviews (please respond to each point):

*Reviewer #1:*

The manuscript by Stefanius et al. reports that exosomes from a tumor cell line can initiate the process of malignant transformation in NIH-3T3 cells, which is a non-malignant stromal cell line. The authors have used exosomes from 4 pancreatic cancer cell lines as well as non-malignant immortalized pancreatic cells as an initiator in the classical cell transformation assay. The authors report that exosomes from malignant cells but not normal cells can act as initiators of cell transformation in the presence of different promoters. In addition, the exosome-transformed cells can form tumors in immunocompromised mice which histologically resemble tumors of mesenchymal origin. Further, the authors perform proteomic and exome profiling on the exosome transformed cells and observe clear differences in the protein profiles of exosome-transformed NIH-3T3 cells when compared to normal cells.

Overall, the data in Figure 1 represent an interesting finding that is not too distant from other similar findings in the exosome field, which have demonstrated that exosomes can be oncogenic in certain situations. However, the larger concern here is two fold. First, I am not convinced that the data and interpretation in Figures 2 and 3 are correct, and may be misleading for the field. Second, without a clear mechanism by which the exosomes do this, it is in the realm of phenomenology and therefore not immediately generalizable outside of this one system. This manuscript would need extensive reworking and many additional experiments to make it appropriate for *eLife*.

Essential revisions:

1) In Figure 1, the authors isolate foci from either MCA/TPA or ExC/TpA and then show that these grow in mice. However, they then compare this to parental 3T3 cells. This is not as useful a control since it is possible that the 3T3 culture contains cells capable of giving rise to tumors, and that what the exosomes did was select for that population. Thus, since the 3T3 cells do form foci (albeit at a smaller rate, as shown in Figure 1C) those should be the proper control, and would allow us to determine if exosome treatment is absolutely essential for tumor formation. In this model, the exosomes are not really initiators in the classic sense of the word, but instead just acting as a selection force.

2) It is unclear whether exosomes from normal pancreatic cells also have tumorigenic potential. In Figure 1C, you show that they give rise to foci at a similar rate to the DMSO control but these are never tested in the mice. This is important for the reasons related above, in that it is not clear whether the cancer cell exosome, per se, is required for tumor initiation rather than selecting for a pre-existing cell in the population.

3) The use of the 3T3 assay, while interesting and informative, does not tell us much about how general this phenomenon is. It is essential that the authors utilize other cells to tell us how unique this situation is to the 3T3. For example, they could use normal pancreas cells as well as other normal cell lines such as additional fibroblasts, mesenchymal stem cells, etc.

4) The proteomic analysis of the foci is not especially informative. The authors state that the wide variation across the samples is due to "random molecular changes across the population of treated NIH/3T3 cells". However, there is no evidence that this is really the case, since widespread variation across the samples could be due to technical artifacts from the proteomic analysis itself. How do you control for this possibility in this analysis? Moreover, the authors state "As expected the NIH/3T3 cells left untreated show no marked global changes in protein content…" yet then state that there were significant changes in the foci analysis. But how do you define what "marked" is in one context and not another in a way that is statistically rigorous? In its current form, I am not convinced this analysis is correct.

In addition, an important control experiment would be to perform the proteomic analysis on the tumors that emerged from the mice instead, since conceivably these might have more of a convergent profile that allows them to grow in the animal. In either case, it is difficult to understand how you can definitively state this is due to random effects of the exosomes without much larger numbers of samples and/or additional proteomic controls to eliminate technical artifacts.

5) In Figure 3B, the number of identified variants (more than 12,000) in each of those samples seems extraordinarily high. Were this rate of genomic alteration seen in the rest of the genome, assuming it is random mutation as the authors postulate, that would indicate that genome wide there would be nearly 600,000 mutations (i.e. if 12,000 mutations are seen in the 2% of the genome that is exonic, then one would expect 50X this number genome wide). This would be extraordinary even for highly mutated cancer such as melanoma. My expectation is that with proper stringency measures and false discovery correction, along with greater number of samples, this number of mutations would be markedly reduced.

Moreover, the authors provide no details about their use of the PLINK algorithm to cluster mutation profiles, calling into question how robust this segregation really is. This is especially concerning given the extremely high mutation burden they are seeing, which can make the interpretation of that data very problematic.

6) The interpretation that the exonic changes they are seeing is due to mismatch repair, as outlined in Figure 3 and Table 1, is not convincing. With a mutation rate this high, as outlined before, by random chance the likelihood of seeing a mutation in a mismatch repair gene is extremely high. In no way does this suggest that the mutation profile they are seeing is due to mismatch repair – the correlation is just not strong enough. The authors would absolutely have to restore mismatch repair proteins (i.e. overexpress it in the exosome treated cells) and demonstrate that this eliminates: 1) the foci formation, 2) mouse tumorigenicity, and 3) sequencing results. Otherwise, without this data, the assumption that exosomes induce random mutations via defects in mismatch repair may not be correct, and could lead the field astray.

7) Finally, the authors posit that some factor from the exosomes is inducing this initiator effect, presumably via changes in mismatch repair or possibly other mechanisms. However, no attempt to identify the mechanism for this effect is made – is it protein, mRNA, RNA, lipid, other? Although I recognize that the identification of the exosome component mediating this effect is a major challenge, without this information it is difficult to really determine the overall importance of these effects. This is an essential experiment, and likely to have major impacts on the interpretation of how exosomes act to induce oncogenic changes in nearby cells.

Additional data files and statistical comments:

The proteomic and exonic analysis is not convincing in its current form.

*Reviewer #2:*

The manuscript by Stefanius and colleagues provides further evidence that cancer cell derived exosomes can initiate events that can lead to the transformation of NIH3T3 cells. Exosomes from cancer cells irrespective of the Kras mutation status can act as initiators that prime the promotors to transform cells and enable tumor formation by the NIH3T3 cells. This is an interesting paper but it is unclear as to its novelty and innovation as presented. The study has potential. Some comments are provided below.

1) While the use of MCA and TPA has been around for a while, it is not clear what the in vivo relevance is at this point. It would more relevant to use nonchemical promoters not generally found in human tissue. This should be attempted. Would the exosomes initiate NIH3T3 cells if they are injected into mice with their pancreas primed using relevant genetic promoters, and induce tumor formation? Such experiments would move the field forward. The basic discovery that exosomes can induce transformation of cells without such promoting agents (TPA) has been published and should be cited. Some studies actually use serum exosomes from cancer patients to illustrate their ability to induce tumor formation by cells.

http://www.pnas.org/content/108/12/4852/

https://www.ncbi.nlm.nih.gov/pubmed/25446899/

https://www.ncbi.nlm.nih.gov/pubmed/28854931/

https://www.ncbi.nlm.nih.gov/pubmed/27179759/

2) It is unclear why foci were used to induce tumors versus the cells by themselves after the exosomes mediated initiation and TPA induced promotion. This should be addressed.

3) The amount of exosomes used is not clear and whether it represents physiologically relevant concentrations. What does 80ng/mL represent in particle number? Also, it is critical to perform dose response experiments in this setting.

4) The exclusive use of NIH3T3 limits the implication of the study. These are transformed cells. This study could include use of other cells of different lineages and just fibroblasts. The use of primary cells would make it relevant to human cancers.

5) More control lines others than just HPDE would make the observations more interesting.

6) Mechanistic insights are minimal in this study considering what has already been published. Important questions remain such as are exosomes initiating the purported events, and why is the identification of the mediators important. Why is the DNA damage and mismatched events induced, and why are these similar to MCA need a more experimental explanation. Gain of function and loss of function experiments are important.

7) The fact that the tumors did not identify any changes is interesting and needs to be further addressed.

*Reviewer #3:*

This is a potentially interesting study that clearly shows that (1) Cancer-derived exosomes effectively initiate malignant transformation in mouse NIH3T3 cells in vitro; (2) The transformed NIH3T3 cells (capable of foci formation) form subQ fibrosarcomas; and (3) not unexpectedly, exosome-induced tumor initiation is related to defective DNA repair, however, the repair defects are random (not caused by one specific exosome-associated oncogene).

The manuscript is well executed, and provides multiple controls, such as additional promoters, transposition of initiator and promoter in two-step carcinogenesis assay, etcetera. However, the study its limited to investigation of NH3T3 transformation and these cells are notoriously easy to transform. Also, the biological relevance of the study is somewhat questionable and critical issues remain unresolved.

First, could a similar process occur in vivo, in mice or humans? If so, an individual with a primary pancreatic tumor should have a higher propensity to form other primary tumor(s). Is that the case? Do such individuals have a chance of exposure to promoters? Second, could this phenomenon be extended to other tumor types (not PDAC)? And last, would exosome-initiated tumors be exclusively fibrosarcomas? Other tumor types? Random tumors? Would PDAC exosomes act as tumor initiating agent in pancreatic epithelium?

These questions could be answered only by using PDAC exosomes in a classical two-step carcinogenesis assay, as opposed to in vitro transformation assay followed by tumorigenesis assay. Similar experiments demonstrating transformation events due to the transfer of exosomes' contents to the bystander cells have already been performed using exposures isolated from the tumor cell lines or from the sera of cancer patients (please see reference below). Most of these experiment are performed as a one step process, which also contradicts the conclusions from the present analysis.

Finally, the potential ability of exosomes to promote tumor growth by virtue of transforming host cells could be tested using fluorescent mice (GFP, RFP etc) as donors or acceptors of exosome-producing NIH-3T3 cells. The appearance of fluorescence tagged cells in a non-tagged exotic tumor formed by exosome-transformed NIH-3T3 cells, whose origin could be ascertained using other fluorescent makers would clearly ascertain such recruitment.

Regrettably, this potentially interesting study is insufficiently novel to be published in *eLife* in its present form, given limited scope of analyses performed and potential low relevance of the results.

In favor of the potential relevance of this study, 18% of co-occurring IPMN and ductal adenocarcinomas were likely independent, suggesting that the carcinoma arose from an independent precursor. By contrast, all colloid carcinomas were likely related to their associated IPMNs. However, to link this process to exosome-dependent tumor initiation, one has to analyze exosomes from patients from IPMNs and their potential capacity to induce pancreatic tumors. (Felsenstein et al). However, the present study is extremely preliminary in nature and does not unequivocally address such a possibility. Further experiments are needed to demonstrate the tumor-initiating capacity of exosomes from the neoplastic foci.

https://www.nature.com/articles/s41419-018-0485-1.pdf (Daie et all, Cell Death and Disease (2018)9:454); https://jeccr.biomedcentral.com/track/pdf/10.1186/s13046-017-0587-0 (Abdouh et al., Journal of Experimental & Clinical Cancer Research (2017) 36:113);

http://www.pnas.org/content/108/12/4852 (Antonyak et al., 2011. PNAS 108 (12) 4852-4857); https://doi.org/10.1016/j.bbrc.2014.07.109 (Lee et al., BBRC Volume 451, Issue 2, 22 August 2014, Pages 295-301

http://cancerres.aacrjournals.org/content/77/21/5808.full-text.pdf (Figueroa et al. Cancer Res. 2017 Nov 1;77(21):5808-5819. doi

https://www.ncbi.nlm.nih.gov/pmc/articles/PMC4254633/pdf/nihms631122.pdf (Melo et al., Cancer Cell. 2014 November 10; 26(5): 707-721)

Additional data files and statistical comments:

The experiments as presented are performed rigorously, with appropriate statistical analysis, however the conclusions that could be drawn are limited and insufficiently novel.

Evaluation of the paper after the first round of revision:

Thank you for submitting your article "Human pancreatic cancer cell exosomes, but not human normal cell exosomes, act as an initiator in cell transformation" for consideration by *eLife*.

While you have performed a few additional experiments that strengthen the manuscript, the reviewers agree that many of the original criticisms remain unaddressed. As you are aware, in this *eLife* trial we are committed to ultimately publishing the paper should you choose to proceed with it. However, all of the reviewer and editorial concerns about the validity of these findings will be published in parallel with the paper.

There remain several major areas of concern:

1) The methods used to isolate the exosomes.

Randy Schekman specifically commented upon the new data as follows:

"I have examined the additional data relevant to the concern I raised about the crude fraction they use to claim an exosome activity. As far as I can tell this consists of one data point with a peak fraction of buoyant membranes assayed at one concentration. This would not pass muster by any standard biochemical analysis. I recommend assays across the peak fraction of exosomal marker proteins and a quantitative titration of activity vs. protein concentration in the peak fraction in comparison to the crude starting material. The expectation is that if this activity tracks with exosomes, the specific activity in their biological assay should have been enriched by the buoyant density separation. Without this it is premature to conclude that the activity is associated with exosomes."

2) The notion that exosomes are initiators.

Several of the reviewers raised concerns about using 3T3 cells as the sole evidence of initiation activity. This was not addressed experimentally, but instead the authors state that "It will be very interesting to test this function in other cell types, and will be done in future studies." Despite this being a major concern, this is still left unaddressed.

Why this is important is related to the new data in Figure 3B, Figure 3—figure supplement 2A, they show that the background foci actually do form tumors, albeit at a slower rate and with lower efficiency than the exosome treated 3T3 foci. But the fact that the background foci do indeed form tumors actually argues against a pure initiator activity – it seems to accelerate what is already happening in the background foci anyway. Thus, a central argument that these exosomes are true initiators does not seem fully accurate. In the absence of other cell types other than 3T3s, it is hard to state this definitively outside of this one context.

3) The proteomic analysis.

While the authors make it clear how they performed the analysis, it is still confusing to state "However, proteins found in transformed cells shown in Figure 4 exhibit more divergence from control cells" Divergence based on what statistical measure? Venn diagrams are not statistical methods, and it is not clear that the PANTHER data shows true statistical enrichment in the exosome treated cells.

4) Exome analysis.

In the rebuttal, the authors state that "we feel confident that the method we used is sufficient to obtain reliable results". Yet the request to analyze this data with additional pipelines and more stringent statistics was not performed. There is no orthogonal validation of this pipeline to ensure that the results are accurate.

In looking at their more detailed methods, it is now clear that they are comparing their exomes to the human reference genomes (GRCm38), so that the variants being called are actually just variants that differ from this reference genome, and not the parental 3T3 cells. How do you know that most of the called variants are not just called SNPs or baseline point mutations between 3T3 cells and the human reference genome? Why not compare the exome data to the parental 3T3 cells rather than the human reference genome?

The authors now provide additional data, using the same pipelines, on bulk exome data on both untreated and TPA treated foci in Figures 5 and 6. In Figure 5—figure supplement 1, they essentially show that the number of mutations in the untreated or TPA treated cells is similar to that in the MCA/TPA or ExC/TPA foci. But if the basic idea is that exosomes act as initiators through increases in mutation rate, then why are the number of mutations essentially the same. Furthermore, in the data in Figure 6, you show that the signature 20 is higher in the MCA/TPA or ExC/TPA samples compared to untreated/TPA, but this just looks like a modest increase, and there is no statistical analysis to say whether these are really different. How do you confirm that you are really enriching for this signature in this dataset other than a heatmap which looks to be so.

Finally, it is not clear that bulk sequencing of this population of highly mutated cells is appropriate. If we consider that each cell could have many different mutations, the more appropriate experiment is to create single cell clones out of the transformed foci, let that grow up, and then perform the exome analysis on these isolated clones. While this is an expensive and ultimately very time consuming method, it would ultimately be much more informative in understanding how the mutations occur in each cell, rather than a population of cells which may or may not have defects is mismatch repair.

5) The role of mismatch repair.

While we recognize that adding in single mismatch repair genes may not rescue the phenotypes, at the same time the authors are making a very bold claim that exosomes specifically lead to MMR defects. This is perhaps the most intriguing idea in the paper, but without any experiments to mechanistically dissect this it remains correlative.

Evaluation of the paper after the second round of revision:

In this revised manuscript, the authors have tried to address several of the issues brought up during the review process. As part of this trial, we will proceed with publication of the manuscript although we do feel several issues remain unresolved that could have further strengthened the paper.

First, Dr Schekman raised a point that dose response relationships comparing the more purified exosomes with the crude starting material would be required to establish that the buoyant density gradient achieved purification of the active vesicle species. The single concentration of the gradient fraction used in the new version of the paper fails to establish meaningful purification.

Second, while we agree that the 3T3 assay is a standard initiator assay, the development of other assays to confirm this potentially new and important function of exosomes would have provided significant confidence that this is a generalizable activity.

Third, the authors have used one informatic pipeline for their analysis, whereas adding in additional and complementary pipelines for mutation calling (i.e MuTect, etc) would have provided increased confidence that these are reproducible results from an informatic standpoint. Analysis of single cell clones from the transformed cells would have also given a more fine-grained analysis of the mutational pattern induced via exosomes, which cannot be achieved by bulk sequencing alone.

---

## [Author Response]

Major concerns:There were several major and significant concerns raised by the reviewers. They are outlined in detail below, but in general there were serious reservations about the informatic analysis and how it was performed, and a concern as to whether the mismatch repair interpretation is robust. More importantly, even if the informatics is corrected, there is no mechanistic insight into how these defects in mismatch repair occur, or even the necessity/sufficiency of this pathway.

Below we have addressed point by point the concerns of each reviewer. Text has been added to explain the informatic analysis to assure the reviewers that there is rigor in all methods used for our conclusions. Additionally, the informatics analysis, particularly the exome sequencing data, has been expanded upon as described in detail below. As mentioned in the title, summation, and manuscript, we have identified, for the first time, that cancer cell exosomes can act as classic initiators in the cell transformation. Our studies provide molecular mechanistic insight by demonstrating that cancer cell exosomes, like other classic chemical initiators, induce random mutations into cells. Cells from this pool of randomly mutated cells can then be driven to transformation by a classic promoter.

After the review was completed, the Editor-in-Chief expressed additional concerns about the conclusions as drawn on the basis of experiments with a crude fraction of particulate material from conditioned medium:Your claims are based on the use of "exosomes" as defined by what you refer to as established protocols for their isolation. Unfortunately, the standards in the field are inadequate and fail to account for many decades of experience in the fractionation of membrane organelles (please see Shurtleff et al., Annual Reviews of Cancer Biology, 2018). The crude particulate fraction obtained by filtration and differential centrifugation of conditioned medium contains other material including contaminating membranes and nonmembranous particles such as large RNPs. Thus, without further fractionation of the relevant sedimentable species, whether membranous or not, your conclusions about the roe of exosomes in the initiation of tumorigenesis will remain suspect.

We agree that one of the current challenges in the exosome field is the lack of a universally accepted purification method. Discussion surrounding this and the current standards used to validate exosome isolation is included in the text. Exosomes were characterized according to guidelines laid out in the Minimal Information for Studies of Extracellular Vesicles (MISEV2018). Additionally, in Figure 2—figure supplement 1B, we show that further fractionation by sucrose density gradient does not change the transformation potential of cancer cell exosomes.

Discussion section:

“To eliminate the possibility that contaminants might participate in the exosome initiator activity, we further purified the exosomes using a sucrose density gradient in order to obtain ‘pure’ exosome fractions (32). Identical experiments with both populations of exosomes demonstrated that both samples could act as an initiator in the CTA, thereby establishing exosomes as the initiating factor in this assay (Figure 2—figure supplement 1B).”

Separate reviews (please respond to each point):

Reviewer #1:

[…] Overall, the data in Figure 1 represent an interesting finding that is not too distant from other similar findings in the exosome field, which have demonstrated that exosomes can be oncogenic in certain situations. However, the larger concern here is two fold. First, I am not convinced that the data and interpretation in Figures 2 and 3 are correct, and may be misleading for the field. Second, without a clear mechanism by which the exosomes do this, it is in the realm of phenomenology and therefore not immediately generalizable outside of this one system. This manuscript would need extensive reworking and many additional experiments to make it appropriate for eLife.Essential revisions:1) In Figure 1, the authors isolate foci from either MCA/TPA or ExC/TpA and then show that these grow in mice. However, they then compare this to parental 3T3 cells. This is not as useful a control since it is possible that the 3T3 culture contains cells capable of giving rise to tumors, and that what the exosomes did was select for that population. Thus, since the 3T3 cells do form foci (albeit at a smaller rate, as shown in Figure 1C) those should be the proper control, and would allow us to determine if exosome treatment is absolutely essential for tumor formation. In this model, the exosomes are not really initiators in the classic sense of the word, but instead just acting as a selection force.

To address this concern, we performed in vivo experiments using background foci from untreated NIH/3T3 cells. Three independent background foci from different wells were isolated, expanded, and injected into NSG (NOD scid γ) mice (n=5).

Subsection “In vivo studies confirm the fully transformed state of cancer cell exosome-initiated cells”:

“After injection of untreated background foci, we observed that six out of 15 total mice formed tumors. Notably, each of the six tumors (5 from the same injected foci) grew later in the time course and at a significantly slower rate compared to initiator-promoter treated transformed foci cells (Figure 3B, Figure 3—figure supplement 2A).”

2) It is unclear whether exosomes from normal pancreatic cells also have tumorigenic potential. In Figure 1C, you show that they give rise to foci at a similar rate to the DMSO control but these are never tested in the mice. This is important for the reasons related above, in that it is not clear whether the cancer cell exosome, per se, is required for tumor initiation rather than selecting for a pre-existing cell in the population.

Quantification of all cell transformation assays shown in Figure 2 and Figure 2—figure supplement 1 consistently show that a uniform level of background foci are formed in all control experiments. This is likely due to the nature of NIH/3T3 cells, as it has been shown that when NIH/3T3 cells are maintained as confluent, non-growing cultures without passage, or injected into mice in high concentrations, they may undergo spontaneous cell transformation (1). Therefore, when stringent focus scoring is carried out according to established criteria (Materials and methods), a constant level of background foci are observed in each experiment. Only in experiments where cells were treated with a functional initiator and promoter, did we observe a significant increase in transformation, resulting in formation of an additional 3 to 4 foci per well. This is why background foci from untreated cells were injected into mice as a control to assess the tumor forming potential of these background foci, as described above.

3) The use of the 3T3 assay, while interesting and informative, does not tell us much about how general this phenomenon is. It is essential that the authors utilize other cells to tell us how unique this situation is to the 3T3. For example, they could use normal pancreas cells as well as other normal cell lines such as additional fibroblasts, mesenchymal stem cells, etc.

Explanation and reasoning behind our use of the classic 2-stage cell transformation assay was added to the Introduction section:

“Malignant transformation of a normal cell occurs in a stepwise fashion. Point mutations in the genome can result in the reprogramming of a normal cell to a less differentiated state that is receptive to additional genetic alterations resulting in uncontrolled growth and ultimately cancer. The classic 2-stage in vitro cell transformation assay (CTA) is a tiered system for transformation that was created for screening potential carcinogenic factors (21-23). In this system, cells are first treated with a suspected carcinogen, called an initiator, such as the genotoxic carcinogen 3-MCA (3-methylcholanthrene). 3-MCA introduces random genetic changes in a pool of normal cells. Subsequently, these initiated cells are exposed to a promoter, such as TPA (12-O-tetradecanoylphorbol 13-acetate), which enhances proliferation in the initiated cells selectively, thus driving malignant transformation of the cells. The resulting transformed cells are observed as foci on a cell culture plate (23, 24). This reductionist approach provides sensitivity in detecting a wider range of initiating agents that may not show obvious transforming activity without a promoter (23). Using this assay as a model system for malignant cell transformation, we assessed whether or not cancer cell-derived exosomes could affect and/or potentially drive the transformation of a normal cell. The results presented herein provide a detailed analysis of a previously unidentified molecular function of cancer cell exosomes for malignant cell transformation.”

It will be very interesting to test this function in other cell types, and will be done in future studies. As the optimization for another assay (2-3 months/assay condition) to be equally reliable as the CTA with 3T3-cells will require over a year, we think this experiment is beyond the scope of this paper.

4) The proteomic analysis of the foci is not especially informative. The authors state that the wide variation across the samples is due to "random molecular changes across the population of treated NIH/3T3 cells". However, there is no evidence that this is really the case, since widespread variation across the samples could be due to technical artifacts from the proteomic analysis itself. How do you control for this possibility in this analysis? Moreover, the authors state "As expected the NIH/3T3 cells left untreated show no marked global changes in protein content…" yet then state that there were significant changes in the foci analysis. But how do you define what "marked" is in one context and not another in a way that is statistically rigorous? In its current form, I am not convinced this analysis is correct.

Specific details pertaining to how the proteomic analysis (Figure 4 and Figure 4—figure supplement 1) are discussed in detail in Materials and methods. Samples were analyzed using a common LC-MS/MS method and appropriate steps were taken to ensure accuracy of results and avoid potential “technical artifacts,” including proper sample prep, instrument calibration, n=3 replicates, PSM filtering <0.01, protein filtering <0.01, etc. The assessment that “no marked global changes in protein content were observed for [initiated] cells treated with MCA or Capan-2 exosomes when compared to proteins found in untreated NIH/3T3 cells” is based on analysis showing that 80% of the total proteins found (2287/2859) were identified in all three conditions, while only 9.7% of the total proteins found (276/2859) were identified in only a single condition (Figure 4—figure supplement 1). However, proteins found in transformed cells shown in Figure 4 exhibit more divergence from control cells. We have altered the presentation of data in Figure 4 to highlight that the differences in protein content between foci result primarily from the contributions of “unique proteins” that are not found in the control cells. To probe these differences further, we performed Gene Ontology enrichment analysis on the proteins found exclusively in transformed foci (Figure 4D).

In addition, an important control experiment would be to perform the proteomic analysis on the tumors that emerged from the mice instead, since conceivably these might have more of a convergent profile that allows them to grow in the animal. In either case, it is difficult to understand how you can definitively state this is due to random effects of the exosomes without much larger numbers of samples and/or additional proteomic controls to eliminate technical artifacts.

We agree that performing proteomic analysis of the tumors that emerged from mice would be an interesting experiment, however we believe it is beyond the scope of this paper. We were primarily interested in determining if any changes could be detected at the protein level as a direct result of either treatment with an initiator for three days or formation of foci from cell transformation. Since we were working in the context of NIH/3T3 cells, each treatment condition could then be compared to background proteins present in untreated NIH/3T3 cells as a control. Additionally, we have changed the language in the text to clarify that we cannot directly attribute changes observed in protein content of transformed foci to random effects of the exosome treatment. We can only compare the protein profiles of NIH/3T3 cells directly after initiation treatment to those of cells after they have transformed.

5) In Figure 3B, the number of identified variants (more than 12,000) in each of those samples seems extraordinarily high. Were this rate of genomic alteration seen in the rest of the genome, assuming it is random mutation as the authors postulate, that would indicate that genome wide there would be nearly 600,000 mutations (i.e. if 12,000 mutations are seen in the 2% of the genome that is exonic, then one would expect 50X this number genome wide). This would be extraordinary even for highly mutated cancer such as melanoma. My expectation is that with proper stringency measures and false discovery correction, along with greater number of samples, this number of mutations would be markedly reduced.Moreover, the authors provide no details about their use of the PLINK algorithm to cluster mutation profiles, calling into question how robust this segregation really is. This is especially concerning given the extremely high mutation burden they are seeing, which can make the interpretation of that data very problematic.

First, we expanded the Materials and methods section 2.6 on exome sequencing analysis to provide more detailed description about our strategy to analyze data obtained during exome sequencing and how all bioinformatics was performed by collaborators and bioinformatic experts, Min S. Kim and Venkat Malladi. We do acknowledge that more stringent criteria (read depth of DP higher than 10) and use of additional programs together with Strelka2 and combining results to obtain consensus from different variant callers could reduce the number of identified variants, but we feel confident that the method we used is sufficient to obtain reliable results.

Second, we do agree with the reviewer’s math, when it pertains to cells not carrying any mutations in genes that would lead to genetic instability at the nucleotide level. In this situation, each round of cell division would not result in an extreme number of new mutations. Based on COSMIC signatures we know that mismatch repair dysfunction is contributing to mutational burden of transformed cells from MCA/TPA and Capan-2/TPA foci leading to possibility where every cell division can amplify the number of mutations exponentially. When analyzing mismatch repair genes in more details, missense mutations were found to be encoded in each of the transformed foci supporting the finding of dysfunctional MMR system. We also see this signature in background foci, although not the top signature which in these cases was COSMIC Signature 3. This signature is associated with failure of DNA double-strand break-repair and would also contribute to a high mutation rate supporting the high number of variants found in all samples.

6) The interpretation that the exonic changes they are seeing is due to mismatch repair, as outlined in Figure 3 and Table 1, is not convincing. With a mutation rate this high, as outlined before, by random chance the likelihood of seeing a mutation in a mismatch repair gene is extremely high. In no way does this suggest that the mutation profile they are seeing is due to mismatch repair – the correlation is just not strong enough. The authors would absolutely have to restore mismatch repair proteins (i.e. overexpress it in the exosome treated cells) and demonstrate that this eliminates: 1) the foci formation, 2) mouse tumorigenicity, and 3) sequencing results. Otherwise, without this data, the assumption that exosomes induce random mutations via defects in mismatch repair may not be correct, and could lead the field astray.

To address this concern we performed additional exome sequencing analysis for three untreated background foci and three TPA- only treated background foci. Figure 6 highlights mutational signatures found in all 12 samples. These additional sequencing results show that COSMIC signature associated with defective DNA mismatch repair, is not the top signature found in the background foci and it gives us more confidence that our result is reliable. In addition, analysis of the mismatch repair genes showed that no mutations were found in three background foci and only one TPA-only focus had mutations, supporting the probability of MMR dysfunction in the transformed foci. We also provide a more detailed explanation on how all bioinformatics was performed in Materials and methods.

Moreover, the interpretation of a COSMIC profile for mismatch repair and genetic instability is valid based on the published methods for analysis of tumor profiles using exosome sequencing.

Ref 37. A. Goncearenco et al., Exploring background mutational processes to decipher cancer genetic heterogeneity. Nucleic Acids Res **45**, W514-w522 (2017).

The experiment suggested by the reviewer to rescue the transformation with wild type copies of mutated genes is not feasible. The type and number of mutations in any one cell caused by mismatch repair and microsatellite instability are large in number and vary from one cell to another. Even cloning cells from a population leads to variation (Orth, K., et al. PNAS USA (1994) 91(20), 9495-9.)

These cells, as they are transformed with multiple mutations, cannot be corrected with the addition of just one gene.

7) Finally, the authors posit that some factor from the exosomes is inducing this initiator effect, presumably via changes in mismatch repair or possibly other mechanisms. However, no attempt to identify the mechanism for this effect is made – is it protein, mRNA, RNA, lipid, other? Although I recognize that the identification of the exosome component mediating this effect is a major challenge, without this information it is difficult to really determine the overall importance of these effects. This is an essential experiment, and likely to have major impacts on the interpretation of how exosomes act to induce oncogenic changes in nearby cells.

The experiments suggested by the reviewer are of interest to us, but would be a major challenge, as stated. It is known that exosomes contain heterogeneous cargo and that no two exosomes contain the same set of (see below). More work on characterizing the exosomes is necessary and we believe this is out of the scope of this specific study.

We have added text to address this issue in the Discussion section:

“A recent study by Felsentein et al. described an interesting finding in which a portion of co-occurring IPMNs in PDAC patients appeared to be genetically unrelated, meaning they shared no mutations in the assayed genes (42). This elicits a fascinating question as to whether cells in a primary tumor could be derived from independent transformation events as opposed to exclusive clonal events (42). Potentially, exosomes secreted by the primary tumor could orchestrate such events. Additional studies towards understanding the cancer cell exosome mediated initiation are needed to address such questions, specifically, investigating the initiation capacity of cancer cell exosomes in more relevant cells like human epithelial cells.”

And:

“Additionally, it is known that exosomes do not uniformly contain the same pool of proteins, lipids, metabolites, or microRNAs, but rather each exosome contains a unique repertoire of biological molecules thus possibly causing a variety of changes in the cells (7, 9, 12). Therefore, potentially both MCA and cancer cell exosomes are causing random molecular changes across the population of treated NIH/3T3 cells that cannot be detected amongst the overall protein composition of cells by mass spectrometry. Ultimately, more studies are needed to elucidate the molecular mechanism of this cancer cell exosome-mediated initiation event.”

Additional data files and statistical comments:The proteomic and exonic analysis is not convincing in its current form.

We have addressed this concern in our Materials and methods and demonstrate rigor in our analyses. Please see above comments specifically regarding proteomic and exome sequencing data analysis.

Reviewer #2:

The manuscript by Stefanius and colleagues provides further evidence that cancer cell derived exosomes can initiate events that can lead to the transformation of NIH3T3 cells. Exosomes from cancer cells irrespective of the Kras mutation status can act as initiators that prime the promotors to transform cells and enable tumor formation by the NIH3T3 cells. This is an interesting paper but it is unclear as to its novelty and innovation as presented. The study has potential. Some comments are provided below.1) While the use of MCA and TPA has been around for a while, it is not clear what the in vivo relevance is at this point. It would more relevant to use nonchemical promoters not generally found in human tissue. This should be attempted. Would the exosomes initiate NIH3T3 cells if they are injected into mice with their pancreas primed using relevant genetic promoters, and induce tumor formation? Such experiments would move the field forward. The basic discovery that exosomes can induce transformation of cells without such promoting agents (TPA) has been published and should be cited. Some studies actually use serum exosomes from cancer patients to illustrate their ability to induce tumor formation by cells.

http://www.pnas.org/content/108/12/4852 https://www.ncbi.nlm.nih.gov/pubmed/25446899/

https://www.ncbi.nlm.nih.gov/pubmed/28854931/https://www.ncbi.nlm.nih.gov/pubmed/27179759/

Explanation and reasoning behind our use of the classic 2-stage cell transformation assay was added to the Introduction section:

“Malignant transformation of a normal cell occurs in a stepwise fashion. Point mutations in the genome can result in the reprogramming of a normal cell to a less differentiated state that is receptive to additional genetic alterations resulting in uncontrolled growth and ultimately cancer. The classic 2-stage in vitro cell transformation assay (CTA) is a tiered system for transformation that was created for screening potential carcinogenic factors (21-23). In this system, cells are first treated with a suspected carcinogen, called an initiator, such as the genotoxic carcinogen 3-MCA (3-methylcholanthrene). 3-MCA introduces random genetic changes in a pool of normal cells. Subsequently, these initiated cells are exposed to a promoter, such as TPA (12-O-tetradecanoylphorbol 13-acetate), which enhances proliferation in the initiated cells selectively, thus driving malignant transformation of the cells. The resulting transformed cells are observed as foci on a cell culture plate (23, 24). This reductionist approach provides sensitivity in detecting a wider range of initiating agents that may not show obvious transforming activity without a promoter (23). Using this assay as a model system for malignant cell transformation, we assessed whether or not cancer cell-derived exosomes could affect and/or potentially drive the transformation of a normal cell. The results presented herein provide a detailed analysis of a previously unidentified molecular function of cancer cell exosomes for malignant cell transformation.”

It will be very interesting to test this function in other cell types, and will be done in future studies. As the optimization for another assay (2-3 months/assay condition) to be equally reliable as the CTA with 3T3-cells will require over a year, we think this experiment is beyond the scope of this paper.

We have cited the suggested papers and incorporated discussion on the role that exosomes have been previously shown to play in cell transformation in the Discussion section:

“Recent studies demonstrate that cancer cell exosomes can in fact participate in cell transformation, however the details on how these exosomes contribute to transformation remained elusive. For example, Dai et al. showed that exosomes participated in the transformation process of normal cells after cells were treated with arsenite (15). Another study described a transfer of malignant traits to BRCA1 deficient human fibroblast when treated with either cancer patient sera or isolated cancer cell exosomes, leading to malignant transformation (16, 17). In addition, exosomes in breast cancer patient sera have been shown to promote nontumorigenic epithelial cells to form tumors in a Dicer-dependent manner (11). In each of the aforementioned studies, non-transformed cells were either pre-exposed to transforming agents or treated with cancer patient sera or medium from cultured cancer cells. Interestingly, Antonyak et al. has demonstrated that sustained treatment of NIH/3T3 cells with breast cancer or glioma cell-derived microvesicles has the ability to induce transformation of recipient cells (18). Contrastingly, in our studies, when cells were treated only with pancreatic cancer cell exosomes, increased cell transformation was not observed. While the specific details and effects on cell transformation attributed to exosomes vary between the studies, both demonstrate that exosomes are indeed playing a role in cell transformation.”

2) It is unclear why foci were used to induce tumors versus the cells by themselves after the exosomes mediated initiation and TPA induced promotion. This should be addressed.

In the 2-stage cell transformation assay, the phenotype for transformation is focus formation, specifically representing the transformed cells among the total cell population. For this reason, foci were used for in vivo assays. This specific issue is added to the Discussion section:

“The classic 2-stage CTA was specifically used as a tool to study cell transformation because it is rigorous, reproducible, and has a well-established phenotype.”

Additionally, we wanted to assess the heterogeneity between foci formed from the same treatment. This is why multiple independent foci of the same type were compared for all treatment conditions. Discussion on the use of foci cells in the in vivo assay is stated in the text in subsection “In vivo studies confirm the fully transformed state of cancer cell exosome-initiated cells”:

“An important step of assessing the tumorigenic property of transformed cells is their ability to form tumors in vivo. To determine whether exosome-initiated transformed cells have the capacity to form tumors when injected subcutaneously into immunocompromised mice, we first isolated and expanded foci cells from the MCA/TPA and Capan-2 exosome/TPA experiments (Figure 3A).”

“Additional in vivo studies were performed to analyze the tumor forming potential of a variety of foci, including background foci formed in control experiments. (Figure 3, Figure 3—figure supplement 1, Figure 3—figure supplement 2).”

3) The amount of exosomes used is not clear and whether it represents physiologically relevant concentrations. What does 80ng/mL represent in particle number? Also, it is critical to perform dose response experiments in this setting.

We agree and to address this we performed nanoparticle tracking analysis and based on this analysis, 80ng/mL is equivalent to 7.0x10^7^ particles/mL (Figure 1, Figure 1—figure supplement 1).

In addition, we have performed dose response experiments and added results to Figure 2—figure supplement 1C. Results are discussed in the text:

“Dose-response studies were performed using protein concentration as a normalization strategy to evaluate the amount of cancer cell exosomes needed to initiate cell transformation. As a standard concentration in each transformation assay we used 80 ng/mL of proteins, corresponding to 7.0x10^7^ particles/mL. Exosome protein concentrations ranging from 0.08 ng/mL to 2400 ng/mL were tested and we observed equal cell transformation for all concentrations with the exception of the two lowest, 0.08 ng/mL and 0.8 ng/mL. This indicates that initiator activity of cancer cell exosomes requires one dose of exosomes over a 3-day period with a protein concentration of at least 8 ng/mL (Figure 2—figure supplement 1C, Figure 2—figure supplement 1-source data 3).”

A reliable estimate of the number of particles made by cancer cells in vivo is not readily available, but based on own data we observe over a two log increase in exosome production from cancer cells compared to normal cells in vitro

4) The exclusive use of NIH3T3 limits the implication of the study. These are transformed cells. This study could include use of other cells of different lineages and just fibroblasts. The use of primary cells would make it relevant to human cancers.

Explanation and reasoning behind our use of the classic 2-stage cell transformation assay was added to the Introduction section:

“Malignant transformation of a normal cell occurs in a stepwise fashion. Point mutations in the genome can result in the reprogramming of a normal cell to a less differentiated state that is receptive to additional genetic alterations resulting in uncontrolled growth and ultimately cancer. The classic 2-stage in vitro cell transformation assay (CTA) is a tiered system for transformation that was created for screening potential carcinogenic factors (21-23). In this system, cells are first treated with a suspected carcinogen, called an initiator, such as the genotoxic carcinogen 3-MCA (3-methylcholanthrene). 3-MCA introduces random genetic changes in a pool of normal cells. Subsequently, these initiated cells are exposed to a promoter, such as TPA (12-O-tetradecanoylphorbol 13-acetate), which enhances proliferation in the initiated cells selectively, thus driving malignant transformation of the cells. The resulting transformed cells are observed as foci on a cell culture plate (23, 24). This reductionist approach provides sensitivity in detecting a wider range of initiating agents that may not show obvious transforming activity without a promoter (23). Using this assay as a model system for malignant cell transformation, we assessed whether or not cancer cell-derived exosomes could affect and/or potentially drive the transformation of a normal cell.

The results presented herein provide a detailed analysis of a previously unidentified molecular function of cancer cell exosomes for malignant cell transformation.”

It will be very interesting to test this function in other cell types, and will be done in future studies. As the optimization for another assay (2-3 months/assay condition) to be equally reliable as the CTA with 3T3-cells will require over a year, we think this experiment is beyond the scope of this paper.

5) More control lines others than just HPDE would make the observations more interesting.

As suggested by the reviewer, we have added an additional “normal cell” exosome control in addition to HPDE cells. Exosomes were isolated from human primary dermal fibroblasts and tested in the cell transformation assay as an initiator. Results show that like HPDE exosomes, exosomes from primary fibroblasts are unable to initiate cell transformation above background levels. Please refer to Figure 2—figure supplement 1A.

6) Mechanistic insights are minimal in this study considering what has already been published. Important questions remain such as are exosomes initiating the purported events, and why is the identification of the mediators important. Why is the DNA damage and mismatched events induced, and why are these similar to MCA need a more experimental explanation. Gain of function and loss of function experiments are important.

First, as mentioned in the title, summation and manuscript, we have identified, for the first time, that cancer cell exosomes can act as classic initiators in the cell transformation. We have cited previous studies and incorporated discussion on the role that exosomes have been previously shown to play in cell transformation in the Discussion. Our studies provide mechanistic insight by demonstrating that cancer cell exosomes, like other classic chemical initiators, induce random mutations into cells. Cells from this pool of randomly mutated cells can then be driven to transformation by a classic promoter.

Second, we also were surprised that MCA/TPA and Capan-2 exosome/TPA transformed foci have similar mutational signatures. To understand this better, we performed additional exome sequencing studies for three untreated background foci and three TPA- only treated background foci to learn whether the same mutational signature is seen in foci without any treatment or treatment with the TPA, as these were the two common factors (NIH-3T3 cells and TPA) with our previous samples. Our exosome analysis revealed that all of the sequenced background foci (6) had a similar mutagenic profile whereas none of the transformed foci encoded a similar mutagenic profile. These additional results show that mismatch repair is a main driver for mutations found in transformed cells and that the COSMIC signature associated with defective DNA mismatch repair is not the top signature found in the background foci. We discuss these findings in the Discussion:

“Moreover, even though all six transformed foci were derived from treatment with a common promoter (TPA), foci from TPA-only treated cells do not show the same mutational signatures as the initiator/promoter transformed foci. This supports a proposal that when TPA is used as a promoter, the molecular path towards tumorigenesis is not random, but rather TPA will drive a path using randomly mutated cells that accommodates a course towards a Cosmic 20 or 15 profile. Furthermore, future studies might reveal that another promoter acting on initiator treated cells might lead towards another common endpoint with regards to the 30 different mutagenic profiles.”

7) The fact that the tumors did not identify any changes is interesting and needs to be further addressed.

As stated in subsection “In vivo studies confirm the fully transformed state of cancer cell exosome-initiated cells”:

“Histological analysis confirmed that the tumors are fibrosarcomas, as was expected because cells used in the CTA are NIH/3T3 cells from a mesenchymal origin (Figure 3—figure supplement 1D).”

Reviewer #3:

This is a potentially interesting study that clearly shows that (1) Cancer-derived exosomes effectively initiate malignant transformation in mouse NIH3T3 cells in vitro; (2) The transformed NIH3T3 cells (capable of foci formation) form subQ fibrosarcomas; and (3) not unexpectedly, exosome-induced tumor initiation is related to defective DNA repair, however, the repair defects are random (not caused by one specific exosome-associated oncogene).

The manuscript is well executed, and provides multiple controls, such as additional promoters, transposition of initiator and promoter in two-step carcinogenesis assay, etcetera. However, the study its limited to investigation of NH3T3 transformation and these cells are notoriously easy to transform. Also, the biological relevance of the study is somewhat questionable and critical issues remain unresolve

First, could a similar process occur in vivo, in mice or humans? If so, an individual with a primary pancreatic tumor should have a higher propensity to form other primary tumor(s). Is that the case?

These are interesting questions raised by the reviewer and we discussed this as a fascinating idea based on a recent study by Felsentein and colleagues where they describe a finding in which a portion of co-occurring IPMNs in PDAC patients appeared to be genetically unrelated, meaning they shared no mutations in the assayed genes (Discussion section):

“A recent study by Felsentein et al. described an interesting finding in which a portion of co-occurring IPMNs in PDAC patients appeared to be genetically unrelated, meaning they shared no mutations in the assayed genes (42). This elicits a fascinating question as to whether cells in a primary tumor could be derived from independent transformation events as opposed to exclusive clonal events (42). Potentially, exosomes secreted by the primary tumor could orchestrate such events. Additional studies towards understanding the cancer cell exosome mediated initiation are needed to address such questions, specifically, investigating the initiation capacity of cancer cell exosomes in more relevant cells like human epithelial cells.”

In addition, to further address questions asked by this reviewer, we have incorporated discussion on the role that exosomes have been previously shown to play in cell transformation in the Discussion section:

“Recent studies demonstrate that cancer cell exosomes can in fact participate in cell transformation, however the details on how these exosomes contribute to transformation remained elusive. For example, Dai et al. showed that exosomes participated in the transformation process of normal cells after cells were treated with arsenite (15). Another study described a transfer of malignant traits to BRCA1 deficient human fibroblast when treated with either cancer patient sera or isolated cancer cell exosomes, leading to malignant transformation (16, 17). In addition, exosomes in breast cancer patient sera have been shown to promote nontumorigenic epithelial cells to form tumors in a Dicer-dependent manner (11). In each of the aforementioned studies, non-transformed cells were either pre-exposed to transforming agents or treated with cancer patient sera or medium from cultured cancer cells. Interestingly, Antonyak et al. has demonstrated that sustained treatment of NIH/3T3 cells with breast cancer or glioma cell-derived microvesicles has the ability to induce transformation of recipient cells (18). Contrastingly, in our studies, when cells were treated only with pancreatic cancer cell exosomes, increased cell transformation was not observed. While the specific details and effects on cell transformation attributed to exosomes vary between the studies, both demonstrate that exosomes are indeed playing a role in cell transformation.”

Do such individuals have a chance of exposure to promoters?

Most likely everyone is exposed to such factors, as many known promoters are made by cells. Normal cells are able to use intact signaling machinery and defense mechanisms to respond to such factors. These would include growth factors, oxidative-stress, inflammation, etc. Both oxidative stress and inflammation are associated with pancreatitis and pancreatic cancer and could function as an intrinsic promoter.

Second, could this phenomenon be extended to other tumor types (not PDAC)? And last, would exosome-initiated tumors be exclusively fibrosarcomas? Other tumor types? Random tumors? Would PDAC exosomes act as tumor initiating agent in pancreatic epithelium?

Explanation and reasoning behind our use of the classic 2-stage cell transformation assay was added to the Introduction section:

“Malignant transformation of a normal cell occurs in a stepwise fashion. Point mutations in the genome can result in the reprogramming of a normal cell to a less differentiated state that is receptive to additional genetic alterations resulting in uncontrolled growth and ultimately cancer. The classic 2-stage in vitro cell transformation assay (CTA) is a tiered system for transformation that was created for screening potential carcinogenic factors (21-23). In this system, cells are first treated with a suspected carcinogen, called an initiator, such as the genotoxic carcinogen 3-MCA (3-methylcholanthrene). 3-MCA introduces random genetic changes in a pool of normal cells. Subsequently, these initiated cells are exposed to a promoter, such as TPA (12-O-tetradecanoylphorbol 13-acetate), which enhances proliferation in the initiated cells selectively, thus driving malignant transformation of the cells. The resulting transformed cells are observed as foci on a cell culture plate (23, 24). This reductionist approach provides sensitivity in detecting a wider range of initiating agents that may not show obvious transforming activity without a promoter (23). Using this assay as a model system for malignant cell transformation, we assessed whether or not cancer cell-derived exosomes could affect and/or potentially drive the transformation of a normal cell.

The results presented herein provide a detailed analysis of a previously unidentified molecular function of cancer cell exosomes for malignant cell transformation.”

It will be very interesting to test this function in other cell types, and will be done in future studies. As the optimization for another assay (2-3 months/assay condition) to be equally reliable as the CTA with 3T3-cells will require over a year, we think this experiment is beyond the scope of this paper.

These questions could be answered only by using PDAC exosomes in a classical two-step carcinogenesis assay, as opposed to in vitro transformation assay followed by tumorigenesis assay. Similar experiments demonstrating transformation events due to the transfer of exosomes' contents to the bystander cells have already been performed using exposures isolated from the tumor cell lines or from the sera of cancer patients (please see reference below). Most of these experiment are performed as a one step process, which also contradicts the conclusions from the present analysis.

Actually, other studies are done in the context of another factor. We discussed this in the Discussion section:

“Recent studies demonstrate that cancer cell exosomes can in fact participate in cell transformation, however the details on how these exosomes contribute to transformation remained elusive. For example, Dai et al. showed that exosomes participated in the transformation process of normal cells after cells were treated with arsenite (15). Another study described a transfer of malignant traits to BRCA1 deficient human fibroblast when treated with either cancer patient sera or isolated cancer cell exosomes, leading to malignant transformation (16, 17). In addition, exosomes in breast cancer patient sera have been shown to promote nontumorigenic epithelial cells to form tumors in a Dicer-dependent manner (11). In each of the aforementioned studies, non-transformed cells were either pre-exposed to transforming agents or treated with cancer patient sera or medium from cultured cancer cells. Interestingly, Antonyak et al. has demonstrated that sustained treatment of NIH/3T3 cells with breast cancer or glioma cell-derived microvesicles has the ability to induce transformation of recipient cells (18). Contrastingly, in our studies, when cells were treated only with pancreatic cancer cell exosomes, increased cell transformation was not observed. While the specific details and effects on cell transformation attributed to exosomes vary between the studies, both demonstrate that exosomes are indeed playing a role in cell transformation.”

Finally, the potential ability of exosomes to promote tumor growth by virtue of transforming host cells could be tested using fluorescent mice (GFP, RFP etc) as donors or acceptors of exosome-producing NIH-3T3 cells. The appearance of fluorescence tagged cells in a non-tagged exotic tumor formed by exosome-transformed NIH-3T3 cells, whose origin could be ascertained using other fluorescent makers would clearly ascertain such recruitment.

We appreciate these suggestions made by the reviewer and will take these into consideration in future studies but we think that they are beyond the scope of this paper.

Regrettably, this potentially interesting study is insufficiently novel to be published in eLife in its present form, given limited scope of analyses performed and potential low relevance of the results.

We specifically used the 3T3 assay to ask a precise mechanistic question because this would allow us to use a reductionist approach to define a very elusive question. Do cancer cell or normal cell exosomes act as an initiator, promoter or both in cell transformation? We utilized the 2-stage CTA as a reliable model assay to answer this question. In addition, the 2-stage CTA could be performed with NIH/3T3 cells using an established chemical initiator (MCA) and promoters (TPA and CdCl_2_).

In addition, as mentioned in the title, summation and manuscript, we have identified, for the first time, that cancer cell exosomes can act as classic initiators in the cell transformation. Our studies provide mechanistic insight by demonstrating that cancer cell exosomes, like other classic chemical initiators, induce random mutations into cells. Cells from this pool of randomly mutated cells can then be driven to transformation by a classic promoter.

In favor of the potential relevance of this study, 18% of co-occurring IPMN and ductal adenocarcinomas were likely independent, suggesting that the carcinoma arose from an independent precursor. By contrast, all colloid carcinomas were likely related to their associated IPMNs. However, to link this process to exosome-dependent tumor initiation, one has to analyze exosomes from patients from IPMNs and their potential capacity to induce pancreatic tumors. (Felsenstein et al). However, the present study is extremely preliminary in nature and does not unequivocally address such a possibility. Further experiments are needed to demonstrate the tumor-initiating capacity of exosomes from the neoplastic foci.https://www.nature.com/articles/s41419-018-0485-1.pdf (Daie et all, Cell Death and Disease (2018)9:454); https://jeccr.biomedcentral.com/track/pdf/10.1186/s13046-017-0587-0 (Abdouh et al., Journal of Experimental & Clinical Cancer Research (2017) 36:113);http://www.pnas.org/content/108/12/4852 (Antonyak et al., 2011. PNAS 108 (12) 4852-4857); https://doi.org/10.1016/j.bbrc.2014.07.109 (Lee et al., BBRC Volume 451, Issue 2, 22 August 2014, Pages 295-301http://cancerres.aacrjournals.org/content/77/21/5808.full-text.pdf (Figueroa et al. Cancer Res. 2017 Nov 1;77(21):5808-5819. doihttps://www.ncbi.nlm.nih.gov/pmc/articles/PMC4254633/pdf/nihms631122.pdf (Melo et al., Cancer Cell. 2014 November 10; 26(5): 707-721)Additional data files and statistical comments:The experiments as presented are performed rigorously, with appropriate statistical analysis…

Thank you for acknowledging the rigor in our work. We have added more detailed information to our Materials and methods to explain our protocols used for bioinformatic analysis.

…however the conclusions that could be drawn are limited and insufficiently novel.

As mentioned in the title, and described in the summation and manuscript, we have identified, for the first time, that cancer cell exosomes can act as classic initiators in the cell transformation. Our studies provide mechanistic insight by demonstrating that cancer cell exosomes, like other classic chemical initiators, induce random mutations into cells. Cells from this pool of randomly mutated cells can then be driven to transformation by a classic promoter.

[Editors’ note: the evaluation of the paper after the first round of revision follows.]

While you have performed a few additional experiments that strengthen the manuscript, the reviewers agree that many of the original criticisms remain unaddressed. As you are aware, in this eLife trial we are committed to ultimately publishing the paper should you choose to proceed with it. However, all of the reviewer and editorial concerns about the validity of these findings will be published in parallel with the paper.There remain several major areas of concern:1) The methods used to isolate the exosomes.Randy Schekman specifically commented upon the new data as follows:"I have examined the additional data relevant to the concern I raised about the crude fraction they use to claim an exosome activity. As far as I can tell this consists of one data point with a peak fraction of buoyant membranes assayed at one concentration. This would not pass muster by any standard biochemical analysis. I recommend assays across the peak fraction of exosomal marker proteins and a quantitative titration of activity vs. protein concentration in the peak fraction in comparison to the crude starting material. The expectation is that if this activity tracks with exosomes, the specific activity in their biological assay should have been enriched by the buoyant density separation. Without this it is premature to conclude that the activity is associated with exosomes.”

The experiments previously requested by this editor in initial reviews were to perform “further fractionation of the relevant sedimentable species” in order to ensure that exosomes were properly isolated and purified from contaminating material. To address this concern, exosomes were further purified on a sucrose density gradient and resulting fractions were blotted for common exosomal markers CD63 and Alix, using the protocol recommended by this editor. Exosome marker proteins were only found in fractions 3 and 4, consistent with previous results. Additionally, nanoparticle tracking analysis of ‘fraction 3’ isolated exosomes clearly show enrichment in particles sized <100nm compared to analysis of ‘crude’ exosomes. After protein concentration of the fractions was determined, Fraction 3 was tested in the cell transformation assay at a concentration of 80 ng/mL (or 7x10^7^ particles/mL) to stay consistent with crude exosome assays. This concentration of exosomes was determined based on dose response studies already performed and described in the paper. When Fraction 3 exosomes were used as an “initiator,” results in Figure 2—figure supplement 1B show that the average foci/well formed is comparable to results from crude exosomes. These experiments demonstrate that both the initial population of exosomes and the further purified fraction of exosomes requested by the editor function as an initiator in the assay.

2) The notion that exosomes are initiators.Several of the reviewers raised concerns about using 3T3 cells as the sole evidence of initiation activity. This was not addressed experimentally, but instead the authors state that "It will be very interesting to test this function in other cell types, and will be done in future studies.” Despite this being a major concern, this is still left unaddressed.

The assay used in this manuscript is a classic assay that is well accepted for discovery of both *initiators* and *promoters* of cellular transformation. This in vitro cell transformation assay (CTA) is proposed as an alternative to in vivo carcinogenicity testing. This assay is reported in the list of accepted methods for REACH (Reg. EC 440/2008). (*1*) Balb/c and NIH/3T3 cells are commonly used cell lines in the Cell Transformation Assay (CTA) and Focus Formation Assay (FFA) (*2*) which are assays that provide a straightforward method to assess the transforming potential of an oncogene or biological component. These assays represent the most well-known CTAs and are regarded as a reliable tool to screen biological components, single chemicals or mixtures for carcinogenicity prediction. In our experiments we used the two-stage cell transformation system which according to Sakai ‘is utilized for measuring tumor promoting activities and also for enhancing the sensitivity of cells to the weak initiators the standard assay is unable to detect’ (*3, 4*). Accordingly, by using this assay the end result produces a reproducible phenotype of foci formation. Rigorous standards for foci scoring have been previously established and were used for our studies (*5*). In this paper, we have used this assay with relevant positive and negative controls and established in a statistically significant and scientific manner that cancer cell exosomes can function as an initiator within the context of the assay.

Unfortunately, the same assay does not result in the same phenotype when using human normal epithelial cell lines. Human cells are known to be more resistant to cell transformation than rodent cells (*6, 7*). This known limitation should not preclude use of the rodent CTA. Currently, there is no other reliable system for the analysis of an *initiation* and *promotion* of cell transformation and it will take significant time to develop a new and reliable 8-week assay. We are in the process of designing a new assay with human cell lines to study the *initiator* and *promoter* activity of known compounds and exosomes, but this new assay is beyond the scope of this study.

As the 3T3 assay is well accepted and rigorous “to screen biological components, single chemicals or mixtures for carcinogenicity prediction”, we think this assay is sufficient for this discovery.

Why this is important is related to the new data in Figure 3B, Figure 3—figure supplement 2A, they show that the background foci actually do form tumors, albeit at a slower rate and with lower efficiency than the exosome treated 3T3 foci. But the fact that the background foci do indeed form tumors actually argues against a pure initiator activity – it seems to accelerate what is already happening in the background foci anyway. Thus, a central argument that these exosomes are true initiators does not seem fully accurate. In the absence of other cell types other than 3T3s, it is hard to state this definitively outside of this one context.

The reviewers initially asked us to test whether background control foci would form tumors when injected into mice. We did these experiments and results showed that background foci from untreated cells could form tumors when injected into mice, albeit these tumors were observed to grow at a less aggressive rate. These background foci are present in all experiments, due to the fact that NIH/3T3 cells are susceptible to transformation. It has been shown that when NIH/3T3 cells are maintained as confluent, non-growing cultures without passage, or injected into mice in high concentrations, they may undergo spontaneous cell transformation (*8*). It is therefore not unexpected that we observed tumor growth after injection of transformed background foci into mice. The probability to spontaneously transform is also increased if cells have been passaged several times (n>4) (*9*). Maintaining the same passaged cells throughout the experiment is an important factor for obtaining reliable results (*9*). Each of our experiments were performed with the sub-confluent cells passaged less than four times.

The presence of these background foci also does not diminish the role of *initiators* and *promoters* in cell transformation. Using strict focus scoring based on previously established criteria, we established a constant level of background foci through all experiments; in each experiment a small number of cells transformed independent of any treatment, as has been observed in previous studies (*10*). This does not argue against the *initiator* activity of exosomes as the reviewer suggests, because only in experiments where cells were treated with a functional *initiator* and a functional *promoter*, did we observe a significant increase in transformation above background. This same increase above background is observed for the established chemical *initiator*, MCA. Finally, in keeping with the definition of an *initiator,* we demonstrate that *initiators* (cancer cell exosomes or MCA) can work with multiple *promotors* (TPA or CdCl_2_).

References:

1. M. G. Mascolo *et al.*, BALB/c 3T3 cell transformation assay for the prediction of carcinogenic potential of chemicals and environmental mixtures. *Toxicology* in vitro*: an international journal published in association with BIBRA***24**, 1292-1300 (2010).

2. in *Encyclopedia of Cancer,* M. Schwab, Ed. (Springer Berlin Heidelberg, Berlin, Heidelberg, 2009), pp. 2081-2081.

3. A. Sakai, BALB/c 3t3 cell transformation assays for the assessment of chemical carcinogenicity. *Alternatives to Animal Testing and Experimentation***14**, 367-373 (2007).

4. A. Sakai, M. Sato, Improvement of carcinogen identification in BALB/3T3 cell transformation by application of a 2-stage method. *Mutation research***214**, 285-296 (1989).

5. K. Sasaki *et al.*, Photo catalogue for the classification of foci in the BALB/c 3T3 cell transformation assay. *Mutation research***744**, 42-53 (2012).

6. T. Akagi, K. Sasai, H. Hanafusa, Refractory nature of normal human diploid fibroblasts with respect to oncogene-mediated transformation. *Proceedings of the National Academy of Sciences of the United States of America***100**, 13567-13572 (2003).

7. S. Creton *et al.*, Cell transformation assays for prediction of carcinogenic potential: state of the science and future research needs. *Mutagenesis***27**, 93-101 (2012).

8. K. Xu, H. Rubin, Cell transformation as aberrant differentiation: Environmentslly dependent spontaneous transformation of NIH 3T3 cells. *Cell Research***1**, 197 (1990).

9. K. Sasaki, A. Huk, N. E. Yamani, N. Tanaka, M. Dusinska, in *Genotoxicity and DNA Repair: A Practical Approach,* L. M. Sierra, I. Gaivão, Eds. (Springer New York, New York, NY, 2014), pp. 343-362.

10. H. Maeshima, K. Ohno, Y. Tanaka-Azuma, S. Nakano, T. Yamada, Identification of tumor promotion marker genes for predicting tumor promoting potential of chemicals in BALB/c 3T3 cells. *Toxicology* in vitro*: an international journal published in association with BIBRA***23**, 148-157 (2009).

3) The proteomic analysis.While the authors make it clear how they performed the analysis, it is still confusing to state "However, proteins found in transformed cells shown in Figure 4 exhibit more divergence from control cells" Divergence based on what statistical measure? Venn diagrams are not statistical methods, and it is not clear that the PANTHER data shows true statistical enrichment in the exosome treated cells.

We have clarified the text pertaining to how the proteomic data is interpreted. Venn diagrams are used to directly compare proteins found in each sample and highlight differences between proteins identified in transformed foci with untreated cells.

4) Exome analysis.In the rebuttal, the authors state that "we feel confident that the method we used is sufficient to obtain reliable results". Yet the request to analyze this data with additional pipelines and more stringent statistics was not performed. There is no orthogonal validation of this pipeline to ensure that the results are accurate.

We used the Strelka2 somatic variation pipeline as previously published and stated in Materials and methods (Kim, S., Scheffler, K. *et al.* (2018) Strelka2: fast and accurate calling of germline and somatic variants. *Nature Methods*, 15, 591-594. doi:10.1038/s41592-018-0051-x). We have clarified the text in the Materials and methods to accurately reflect this in the last three paragraphs of the Materials and methods section.

In looking at their more detailed methods, it is now clear that they are comparing their exomes to the human reference genomes (GRCm38), so that the variants being called are actually just variants that differ from this reference genome, and not the parental 3T3 cells. How do you know that most of the called variants are not just called SNPs or baseline point mutations between 3T3 cells and the human reference genome? Why not compare the exome data to the parental 3T3 cells rather than the human reference genome?

The sequencing data is being compared to mutations found in the parental NIH/3T3 cells. As NIH/3T3 are mouse cells, we compare our data with both the mouse reference genome and then with the original NIH/3T3 cells. As described in Materials and methods, we first align the raw sequencing data to the mouse reference genome (GRCm38). We have clarified the text in the Materials and methods to accurately reflect this in the last three paragraphs of the Materials and methods section.

The authors now provide additional data, using the same pipelines, on bulk exome data on both untreated and TPA treated foci in Figures 5 and 6. In Figure 5—figure supplement 1, they essentially show that the number of mutations in the untreated or TPA treated cells is similar to that in the MCA/TPA or ExC/TPA foci. But if the basic idea is that exosomes act as initiators through increases in mutation rate, then why are the number of mutations essentially the same.

The last statement is incorrect as it compares two unrelated events. The first part of this statement, “But if the basic idea is that exosomes act as initiators through increases in mutation rate”, is correct and we propose that cancer cell exosomes like other *initiators* (such as MCA) act by causing mutations in the *initiator* step of the transformation process. This provides an increase in the number of mutated cells that can be used by a *promoter* to produce more foci than are observed in control wells.

The second part of the statement “then why are the number of mutations essentially the same” is unrelated to the first. The number of mutations in the cells from foci is due to the transformation process whereby a normal cell becomes a transformed cell that has accumulated many mutations of over time. For all the foci, COSMIC mutational profiles (20,15 or 3) involving some type of DNA damage were revealed by MutaGene analysis. As a result, the number of mutations is expected to be large. Additionally, the number of mutations depicted in the Venn diagrams reflects the full set of variants found in each sample, prior to filtering for loss of function or missense mutations. Further filtering of the mutations for loss of function and missense mutations was used when analyzing the specific mutations found in the MMR associated genes (Figure 6—figure supplement 1 and 2) and the 190 oncogenes (Figure 5—figure supplement 1-source data 1).

Furthermore, in the data in Figure 6, you show that the signature 20 is higher in the MCA/TPA or ExC/TPA samples compared to untreated/TPA, but this just looks like a modest increase, and there is no statistical analysis to say whether these are really different.

The data in Figure 6 represents the relative contribution of the pan-cancer derived COSMIC signatures to the mutational profiles of each foci. Contribution of signatures was calculated using MutaGene, the query profile is decomposed into the precomputed sets for COSMIC to determine the exposure of a given sample to different mutation processes (Figure 6—source data 1). Discussion of these profiles is found in the Discussion.

How do you confirm that you are really enriching for this signature in this dataset other than a heatmap which looks to be so.

Respectfully, this is a correct assessment as we cannot definitively confirm or prove the hypothesis that TPA when acting on *initiated* cells will cause mutations in mismatch repair genes. We therefore have changed the text to read:

“These observations support the proposal that when TPA is used as a promoter (to drive proliferation) in combination with a functional initiator (an entity that creates a population of randomly mutated cells), the molecular path of cells towards tumorigenesis may not be random. Rather, TPA may drive a path in a subset of randomly mutated cells that accommodates a course towards COSMIC signatures 20 or 15. Furthermore, future studies might reveal that another promoter acting on initiator treated cells might lead cells towards another common endpoint with regards to the 30 different signature profiles.”

Finally, it is not clear that bulk sequencing of this population of highly mutated cells is appropriate. If we consider that each cell could have many different mutations, the more appropriate experiment is to create single cell clones out of the transformed foci, let that grow up, and then perform the exome analysis on these isolated clones. While this is an expensive and ultimately very time consuming method, it would ultimately be much more informative in understanding how the mutations occur in each cell, rather than a population of cells which may or may not have defects is mismatch repair.

One cannot expect to have a clonal population when a genome encodes a mutation in a mismatch repair gene. This was previously demonstrated:

Orth, K., Hung, J., Gazdar, A., Mathis, M., Bowcock, A., & Sambrook, J. Ovarian tumors display persistent microsatellite instability caused by mutation in the mismatch repair gene hMSH-2. *Cold Spring Harbor Symp. Quant. Biology* (1994) 59, 349-56.

5) The role of mismatch repair.While we recognize that adding in single mismatch repair genes may not rescue the phenotypes, at the same time the authors are making a very bold claim that exosomes specifically lead to MMR defects. This is perhaps the most intriguing idea in the paper, but without any experiments to mechanistically dissect this it remains correlative.

We do not claim that exosomes specifically lead to mismatch repair defects.

We stated that an *initiator* (that causes random mutations) in combination with the *promoter* TPA was observed to result in a mismatch repair phenotype as indicated by our MutaGene analysis.